# Geographical variation in the diatom communities associated with loggerhead sea turtles (*Caretta caretta*)

Bart Van de Vijver[1,2], Käthe Robert[1,2], Roksana Majewska[3,4], Thomas A. Frankovich[5], Aliki Panagopoulou[6], Sunčica Bosak[7]*

1 Meise Botanic Garden, Research Department, Meise, Belgium, 2 Department of Biology—ECOBE, University of Antwerp, Antwerpen, Belgium, 3 Unit for Environmental Sciences and Management, School of Biological Sciences, North-West University, Potchefstroom, South Africa, 4 South African Institute for Aquatic Biodiversity (SAIAB), Grahamstown, South Africa, 5 Institute of Environment, Florida International University, Miami, Florida, United States of America, 6 ARCHELON, The Sea Turtle Protection Society of Greece, Athens, Greece, 7 Department of Biology, Faculty of Science, University of Zagreb, Zagreb, Croatia

* suncica.bosak@biol.pmf.hr

**Data Availability Statement:** All relevant data are within the manuscript and its Supporting Information files.

## Abstract

Epizoic diatoms form an important part of micro-epibiota of marine vertebrates such as whales and sea turtles. The present study explores and compares the diversity and biogeography of diatom communities growing on the skin and shell of loggerhead sea turtles (*Caretta caretta*) from four different localities: Adriatic Sea (Croatia), Ionian Sea (Greece), South Africa and Florida Bay (USA) using both light and scanning electron microscopy. We observed almost 400 diatom taxa belonging to more than 100 genera. Diatom communities from Greece and Croatia showed the highest similarity and were statistically different from those recorded from South Africa and Florida. Part of this variation could be attributed to differences in sampling techniques; however, we believe that geography had an important role. In general, contrary to several previous observations from sea turtles, the presumably exclusively epizoic diatoms contributed less than common benthic taxa to the total diatom flora, which might have been related to the loggerhead feeding behavior. Moreover, skin samples differed from carapace samples in having a distinct diatom composition with a higher proportion of the putative true epizoonts. Our results indicate that epizoic diatom communities differ according to loggerhead geographical location and substrate (skin vs. carapace). The relative abundances of common benthic diatoms and putative exclusive epizoic taxa may inform about sea turtle habitat use or behavior though detailed comparisons among different host species have yet to be performed.

## Introduction

Diatoms (Bacillariophyceae) are unicellular eukaryotic microalgae characterized by a silica outer shell (frustule). The number of diatom species worldwide is estimated between 30 000 and 100 000 [1]. Of these, around 55 000 are estimated to exist in marine habitats. To date,

**Funding:** SB was funded by Croatian Science Fund (HRZZ) UIP-2017-05-5635, RM was funded by the Systematics Association (UK) through the Systematics Research Fund Award (R. Majewska/ 2017). The funders had no role in study design, data collection and analysis, decision to publish, or preparation of the manuscript.

**Competing interests:** The authors have declared that no competing interests exist.

however, less than 5 000 of marine diatoms have been described [2]. Diatoms occur wherever water is available, including terrestrial, freshwater and marine habitats. They are part of the phytobenthos attached to humid or submerged surfaces or thrive as free-floating phytoplankton in open water bodies [3]. Attached diatom communities can be classified by the substratum they live on. For instance, epipsammic diatoms are attached to sand grains, epilithic diatoms grow on rocks, epiphytic diatoms live on plants and epizoic diatoms grow on animals, the latter two categories commonly called epibionts [3].

The external surfaces of large marine vertebrates, such as whales, sea turtles and manatees, provide suitable hard substrata for the development of rich microbial biofilms. In these biofilms, composed of, among others, bacteria, fungi, cyanobacteria, and micro- and macroalgae, diatoms are often one of the main components, with densities sometimes exceeding those known from other marine substrata [4].

Several presumably exclusively epizoic diatom genera including *Bennettella*, *Epipellis*, *Epiphalaina*, *Plumosigma*, and *Tursiocola* have been described from the skin of cetaceans [5–9]. More recently, epizoic diatoms, including novel species, were described from freshwater turtles in the Rio Negro, Brazil [10–11]. Since 2015, there is a growing literature on epizoic diatoms observed on the carapaces and skin of all known sea turtle species [12–15].

Exclusive epibiosis is still debated as a lot of diatom taxa can be found on both animal and non-animal surfaces, and occur only haphazardly on marine turtles as a result of the physical contact with a variety of immersed substrata during the animal feeding and grooming activities [14]. However, currently, several sea turtle-associated genera are considered strictly epizoic. *Chelonicola* and *Poulinea* have so far been found on the carapaces of olive ridleys (*Lepidochelys olivacea*) [13, 14], and later on, green turtles (*Chelonia mydas*) [16–18], flatbacks (*Natator depressus*) [14], hawksbills (*Eretmochelys imbricata*) [14], loggerheads (*Caretta caretta*) [19], and Kemp's ridleys (*Lepidochelys kempii*) [20, 21], whereas *Medlinella* is known only from the skin of loggerheads [12]. Additionally, several new species belonging to non-strictly epizoic genera were described in the recent past from the carapaces of sea turtles. Examples include *Achnanthes elongata* Majewska & Van de Vijver and *A. squaliformis* Majewska & Van de Vijver, found on the carapaces of olive ridleys [17], Kemp's ridleys, loggerheads, and green turtles [20], *Labellicula lecohuiana* Majewska & Van de Vijver, living on the carapaces of green turtles [22, 23], and five species of *Proschkinia*, found associated with different sea turtle species [19].

The present research was conducted on loggerhead sea turtles, named after their large head and jaws. These middle-sized sea turtles (60–200 kg) are characterized in having a yellow coloured plastron and a dark brown carapace. Loggerheads are widely distributed in the subtropical and temperate regions of the Atlantic, Indian and Pacific Ocean and the Mediterranean Sea [24]. They can occur in both deeper areas and shallow river estuaries [25] and are highly migratory. Wallace et al. [24] proposed to subdivide the world loggerhead population into several Regional Management Units (RMUs) that enables the identification of important geographic areas for different subpopulations in terms of their presence, density and richness, including for example Northeast Atlantic, Northwest Atlantic, Mediterranean and Southwest Indian RMU. The present study reports on the diatom communities growing on loggerhead sea turtles from four distinct geographical localities (Adriatic Sea, Ionian Sea (both Mediterranean population), South Africa (Southwest Indian population) and Florida Bay (Northwest Atlantic population) with the objective to provide the baseline data on their diversity, species composition, and biogeography. Additionally, differences between communities living on the various sea turtle body parts (skin versus carapace) are explored.

## Material and methods

### Study area

Samples used in this study were collected from loggerheads found in four different localities: northeastern Adriatic Sea (Croatia), Amvrakikos Gulf (Greece), Kosi Bay (South Africa) and Florida Bay (USA) (Fig 1).

The Adriatic Sea, connected to the Mediterranean Sea via the Otranto Strait, is one of the most important foraging areas for juvenile and adult loggerhead turtles in the Mediterranean Basin [26]. Samples from Adriatic Sea loggerheads were obtained from animals brought into the Marine Turtle Rescue Centre in Aquarium Pula (Croatia) for rehabilitation in 2016 and 2017. A second Mediterranean loggerhead turtle population was sampled in Amvrakikos Gulf (Ionian Sea, Greece), an important foraging area with a very high density of loggerheads [27]. Diatom samples were collected in 2018 in the framework of the research and conservation activities conducted by ARCHELON in Amvrakikos Gulf by capturing them with the rodeo technique [28]. The rodeo technique was also used to capture and sample loggerheads in 2015 during an annual survey of sea turtle populations in Florida Bay, a shallow lagoon. The South

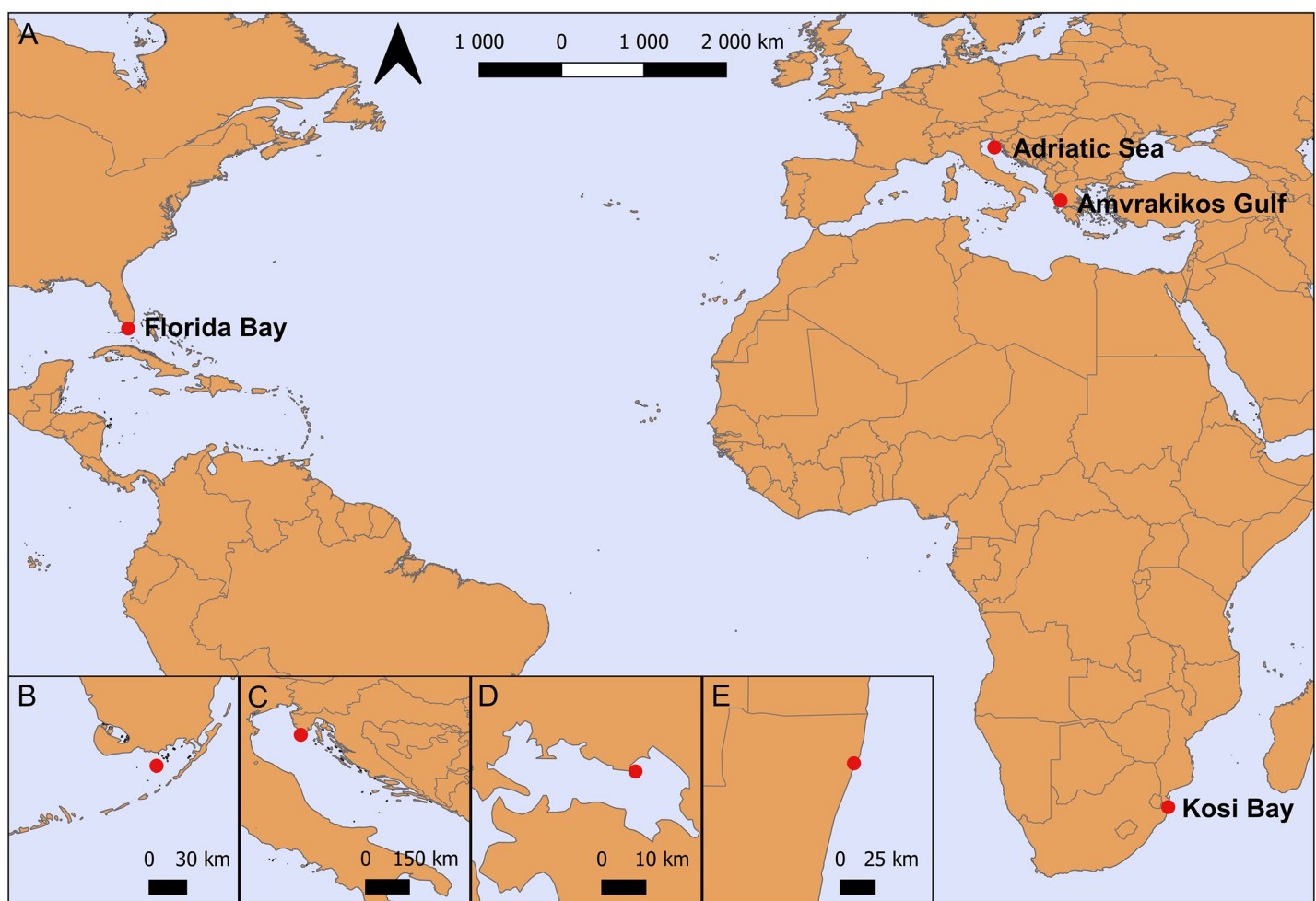

**Fig 1. The sampling areas of loggerhead sea turtles.** (A) Red dots indicate locations of sampled loggerheads. Inserts show details of the sampling locations: (B) Amvrakikos Gulf, Greece; (C) Adriatic Sea, Croatia; (D) Florida Bay, USA; (E) Kosi Bay, South Africa. The maps were made with Natural Earth. Free vector and raster map data @ naturalearthdata.com.

African turtle population was sampled from the beaches in Kosi Bay (northeastern South Africa), an important nesting area for Indian Ocean loggerheads and leatherbacks. Samples were taken in 2018 from nesting loggerheads.

## Sample collection and processing

From each subpopulation, five loggerheads were arbitrarily selected for diatom sampling. Basic information about each turtle, such as carapace length and weight, was also collected at the time of sampling (Table 1). The material collection was performed by researchers licenced for animal handling and well-informed volunteers following institutional guidelines for the care and use of animals. All the procedures involved respecting the ethical standards in the Helsinki Declaration of 1975, as revised in 2013, as well as the applicable national laws. All sampling activities performed in the iSimangaliso Wetland Park (South Africa) were carried out under research permits issued by the South African Department of Environmental Affairs (RES2017/73). Sampling activities in Croatia were done in accordance with the authorization of the Marine Turtle Rescue Centre by the Ministry of Environment and Energy of the Republic of Croatia. Sampling activities in Greece were carried out with permission from the Hellenic Ministry of Agriculture and Environment.

The sampling method differed between sampling events. Carapace samples from Greece and South Africa were collected by scrubbing the carapace with a single-use toothbrush on at least three arbitrarily chosen areas of the carapace, ensuring a scraped surface of at least 60 cm$^2$. Samples were stored in plastic vials filled with at least 70% ethanol for fixation. Carapace samples from Croatia were scraped off with a curette and stored in plastic vials (100 mL) filled

**Table 1. List of samples and information on loggerhead sea turtles.**

| Sample code | Body part | ID Tag / Turtle name | Sampling Date | Sex | Weight (kg) | SCL (cm) | CCL (cm) | SCW (cm) | CCW (cm) |
|---|---|---|---|---|---|---|---|---|---|
| Sampling location: Amvrakikos Gulf (Greece) 39˚ 1' 29" N - 39˚ 1' 44" N; 21˚ 3' 36" E - 21˚ 4' 19" E | | | | | | | | | |
| GRE-01 | carapace | Y6343- Y6344 | 8/1/2018 | | | 75.4 | 78.6 | 55.0 | 68.5 |
| GRE-02 | skin | Y6343- Y6344 | | | | | | | |
| GRE-03 | carapace | Y6366- Y6367 | 8/1/2018 | | | 47.4 | 51.0 | 36.3 | 46.8 |
| GRE-04 | skin | Y6366- Y6367 | | | | | | | |
| GRE-05 | carapace | Y6368- Y6369 | 8/1/2018 | | | 66.1 | 69.6 | 51.5 | 63.8 |
| GRE-06 | skin | Y6368- Y6369 | | | | | | | |
| GRE-07 | carapace | Y6370- Y6371 | 8/1/2018 | | | 54.5 | 58.5 | 43.7 | 55.2 |
| GRE-08 | skin | Y6370- Y6371 | | | | | | | |
| GRE-09 | carapace | M9123- M9124 | 8/1/2018 | | | 50.4 | 53.2 | 37.5 | 47.5 |
| GRE-10 | skin | M9123- M9124 | | | | | | | |
| Sampling location: Florida Bay (USA) 24˚ 55'18" N; 80˚ 48' 28" W | | | | | | | | | |
| FLO-U8 | carapace | HA5053 - HA5054 | 6/24/2015 | F | 58.5 | 74.1 | | | |
| FLO-U9 | carapace | HB5559 - HB5560 | 6/24/2015 | | 48.0 | 66.9 | | | |
| FLO-U10 | carapace | HB5668 - X7596 | 6/24/2015 | M | 89.0 | 87.3 | | | |
| FLO-U11 | carapace | W1924 - W2176 | 6/24/2015 | F | 75.7 | 79.5 | | | |
| FLO-U12 | carapace | HB5581 - HB5582 | 6/25/2015 | | 71.2 | 78.5 | | | |
| Sampling location: Kosi Bay (South Africa) 26˚ 59' 38.9" S; 32˚ 51' 59.8" E | | | | | | | | | |
| SA-33 | carapace | ZA0447D - ZA0427D | 1/16/2018 | F | | | 86.4 | | 83.6 |
| SA-37 | carapace | ZA0829D - ZA0828D | 1/11/2018 | F | | 80.2 | 85.7 | 62.0 | 78.8 |
| SA-45 | carapace | ZA0924D - ZA0186D | 1/11/2018 | F | | 73.9 | 80.2 | 58.2 | 74.8 |

CCL = curved carapace length, CCW = curved carapace width, SCL = straight carapace length, SCW = straight carapace width.

with seawater and fixed with formaldehyde at a final concentration of 4%. Samples from the carapaces of Florida loggerheads were collected using cotton-tipped applicators to rub diatoms from the carapace and onto the cotton tips. The cotton tips were removed from the applicators and stored in sealed plastic bags on ice until further processing. Additionally, diatoms from the skin of loggerheads from Greece were collected by gently scrubbing the dorsal area of the neck and/or the upper side of the flippers with a single-use toothbrush, and then rinsing the toothbrush head into a 50 ml Falcon tube filled with 96% ethanol. In total, we collected 25 samples of which 20 from loggerhead carapace and for five of these animals we were also able to simultaneously sample their skin.

Samples were processed following the methods described by Hasle and Syvertsen [29] for South African samples and van der Werff [30]. In most cases, portions of the biofilm were cleaned by adding 37% $H_2O_2$ and heating to 80˚C for about 1h. The reaction was completed by the addition of $KMnO_4$ [30]. South African samples were digested with boiling concentrated acids ($HNO_3$ and $H_2SO_4$) [29]. Following digestion and centrifugation (three times 10 minutes at 3 500 rpm, Phoenix Instruments, Clinical Centrifuge CD-0412), cleaned material was diluted with distilled water to avoid excessive concentrations of diatom valves on the slides. Samples from Florida were prepared by removing in the laboratory the cotton tips of the applicators using a razor blade and then oxidizing the tips for diatom examination by boiling the cotton fibers of the applicator tip and epizoic organic material in 100 ml of 30% nitric acid followed by addition of potassium dichromate when 50 ml of acid remained. Cleaned diatoms were settled from the mixture for a minimum of 6 h and the remaining acid solution decanted. Settled diatoms were rinsed with deionized water. The rinsing/settling/decanting process was repeated six times until the solution reached a neutral pH. All slides were prepared using Naphrax mounting medium and analyzed using an Olympus BX53 microscope equipped with differential interference contrast (Nomarski) optics and the Olympus UC30 Imaging System. For scanning electron microscope (SEM) analyses, parts of the oxidized suspensions were filtered through a 2 μm Isopore™ polycarbonate membrane filter (Merck Millipore). The filters were mounted on stubs and sputter-coated with 10 nm of platinum or 20 nm of gold-palladium. The stubs were analyzed at Meise Botanic Garden using a JSM-7100F Jeol Field Emission Scanning Electron Microscope at 2 kV and with a working distance of 4 mm. For a more detailed analysis of very finely structured species, some samples were studied using a ZEISS Ultra Scanning Electron Microscope at 3 kV in the Natural History Museum London, UK. Samples and slides are stored at the BR collection (Meise Botanic Garden, Belgium).

In each slide, 400 diatom valves were counted and identified in random transects to estimate the species richness and composition in the samples. After counting, a complete slide was examined to record all occurring taxa in a sample. Extensive literature including both monographs [31–34] and other taxonomic publications were used to identify the observed taxa listed in S1 Table.

## Data analyses

To make the pair-wise comparison between geographic localities we used the Sørenson similarity index [35]. This index uses presence/absence data, and the following formula 2c/(a + b + 2c), where a and b are the numbers of taxa exclusively observed in each of the two populations and c is the number of taxa shared by these populations. The Shannon-Wiener diversity index (ln-based) was calculated using the Multivariate Statistical Package (MVSP) [36]. Abundance data were square-root transformed to downweight dominant taxa. Only taxa with a total abundance of at least 2% in one sample were included in all further statistical analyses to avoid excessive noise in the dataset. Two-dimensional non-metric multidimensional scaling

(NMDS) was used based on Bray-Curtis similarity matrix to reveal the patterns in taxa composition between different localities and turtle body parts. Analysis of similarity percentages (SIMPER) was performed to detect taxa that were responsible for most of the dissimilarity observed between different loggerhead localities and body parts [37]. Two sampling designs, one using four distinct loggerhead subpopulations and the second using two body parts (skin and carapace) were used to perform distance-based permutational multivariate analysis of variance (PERMANOVA) [38]. The PERMANOVA pairwise test was performed on the matrix of square root transformed data calculated on Bray Curtis similarity, using Type III Sums of Squares (i.e. partial sums of squares), with fixed effects and unrestricted permutation of raw data (9999 permutations). All multivariate analyses were performed using the software packages PRIMER v6 and v7 [39], including the add-on package v6 PERMANOVA+.

## Results

### Taxonomic composition and diversity

A total of 183 diatom taxa (including species, varieties and forms) belonging to 56 genera were identified during the counts. One carapace sample from Florida (sample FLO-U8) did not contain a sufficient number of diatom valves and was therefore removed from further analyses. An additional 214 taxa were observed outside the count procedures, bringing the total number of recorded taxa to 397 (S2 Table). Several common diatoms found on both carapace and skin of loggerheads from all four investigated localities are illustrated in Fig 2. Only 41% (166 taxa) of all taxa could be identified to the species level. An additional 14% (56) were given provisional names as 'cf.'. In the Florida and Greek samples, more taxa were identified at the species level (57% and 52%, respectively) compared to the Croatian and South African samples (40% and 37%, respectively).

The most taxon-rich genera found in all samples included *Mastogloia* (42 taxa), *Navicula* (32 taxa), *Amphora* (30 taxa) and *Nitzschia* (30 taxa) (Table 2). Diatom genus composition differed among the carapace samples from different locations. The carapace flora on loggerheads sampled in Croatia contained mostly *Nitzschia* (13 taxa) and *Mastogloia* (11 taxa) whereas in Greek samples *Navicula* (13 taxa) and *Mastogloia* (12 taxa), in Florida samples *Mastogloia* (26 taxa) and the South African samples *Cocconeis* (16 taxa) and *Licmophora* (15 taxa) were the most species-rich genera.

Diatom counts indicated that the most frequently occurring species in all (carapace + skin) samples were *Nitzschia* CRO sp.2 (present 83.3% of all samples), *Amphora crenulata* Wachn. & E.E.Gaiser (70.8%), *Cocconeis lineata* Ehrenb. (70.8%), *Nitzschia* cf. *inconspicua* (62.5%) and *Poulinea* CRO sp.1 (54.2%). Of all counted valves, *N.* cf. *inconspicua* contributed to 16.6%, *Hyalosynedra laevigata* (Grunow) D.M.Williams & Round to 13.1%, *Nitzschia* CRO sp.2 (12.2%), *Chelonicola* SA sp.1 (9.9%), and *Poulinea* CRO sp.1 (4.4%). Altogether, only ten taxa contributed more than 71% of all counted valves, whereas 17 taxa together account for 1% of the total number of valves.

Although most taxa occurred in only one investigated locality, the Greek and Croatian samples shared 45 taxa, a relatively large number (Fig 3, S2 Table). Several taxa, such as *Nitzschia* cf. *inconspicua* and *Nitzschia* CRO sp. 2, appeared abundantly in almost all groups except the Florida samples. In the Florida samples, taxa such as *Hyalosynedra laevigata*, *Synedra bacillaris* (Grunow) Hust. and *Toxarium hennedyanum* (W.Greg.) Pelletan reached the highest relative abundances (Fig 3). All South African samples were dominated by *Chelonicola* SA sp.1, whereas those collected from the Mediterranean region (the Croatian-Greek group) by *Poulinea* CRO sp.1 and sp. 2., *Amphora crenulata*, *Berkeleya fennica* Juhl.-Dannf., *Cocconeis lineata*, and *Navicula* cf. *perminuta* Grunow (Fig 3). One-fifth of all counted valves belonged to the

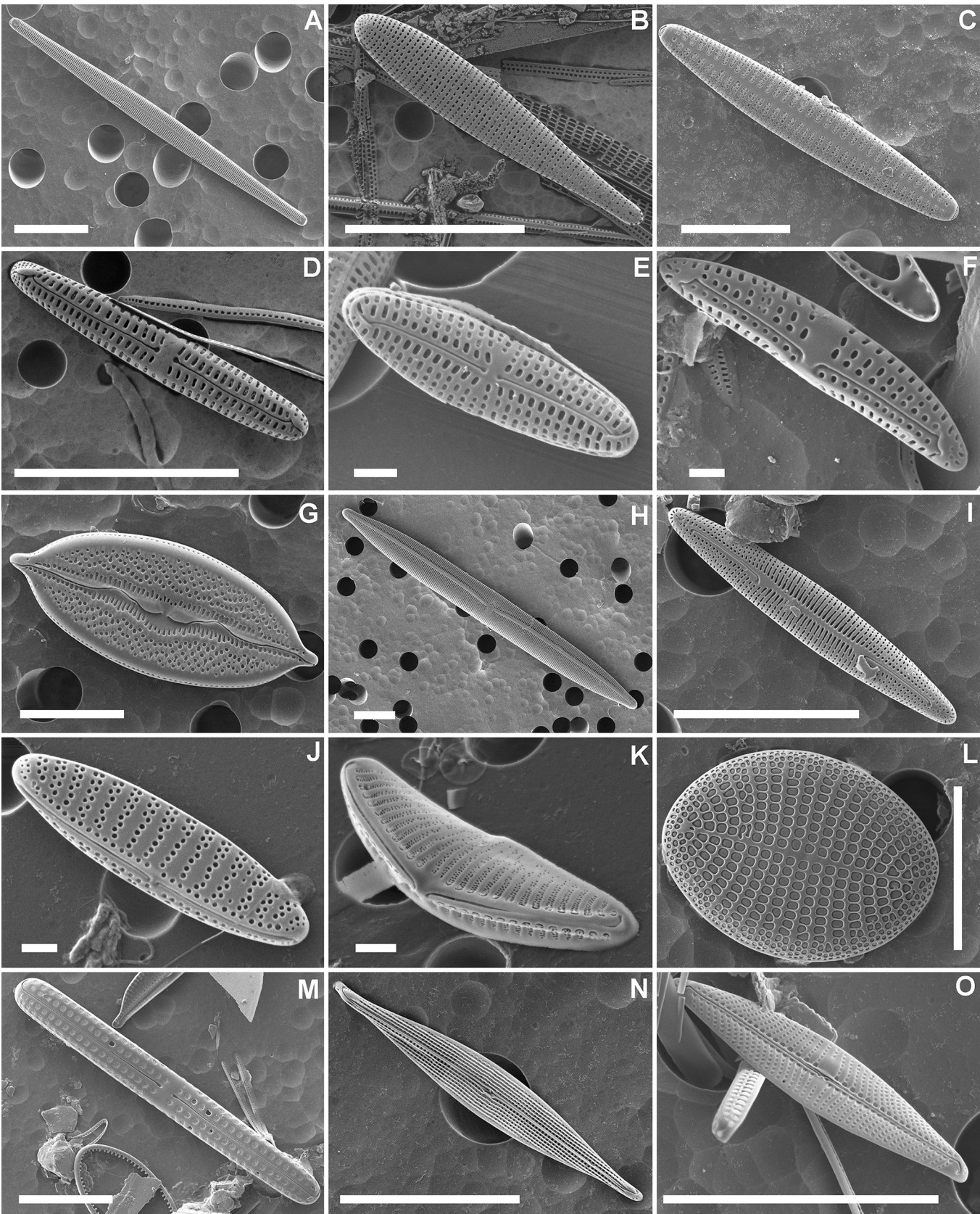

**Fig 2. Scanning electron micrographs of diatom taxa associated with loggerhead sea turtles.** (A) *Hyalosynedra laevigata* (FLO). (B) *Licmophora debilis* (GRE). (C) *Tabularia* cf. *investiens* (FLO). (D) *Poulinea lepidochelicola* (CRO). (E) *Chelonicola* sp. (SA) (F) *Medlinella amphoroidea* (GRE). (G) *Mastogloia* cf. *corsicana* (FLO). (H) *Nitzschia* cf. *scalpelliformis* (FLO). (I) *Berkeleya fennica* (CRO). (J) *Nitzschia* cf. *inconspicua* (CRO). (K) *Bifibulatia* sp. (CRO) (L) *Cocconeis scutellum* (GRE). (M) *Achnanthes elongata* (GRE). (N) *Proschkinia sulcata* (GRE) (O) *Proschkinia vergostriata* (GRE). Scale bars represent 10 μm, except for E, F, J & K where scale bar = 1 μm. CRO–Croatia, Adriatic Sea; GRE = Greece, Ionian Sea; FLO = Florida Bay, USA; SA–South Africa; Kosi Bay.

presumed exclusively-epizoic taxa such as *Achnanthes elongata* Majewska & Van de Vijver, *Medlinella amphoroidea* Frankovich et al., *Poulinea* spp., and *Chelonicola* sp. (S2 Table). The total relative abundance of these species varied strongly among the populations (Fig 3), with the lowest values (0.3%) recorded from the Florida samples and highest from the South African ones (49.1%). Moreover, a significant difference was observed between the carapace and skin samples from Greece, where the relative abundance of the presumably epizoic taxa reached 25.7% and 5.3%, respectively.

Species number in a single sample varied between 11 and 111 taxa (including taxa observed outside the counts) (Fig 4A), and the median from the same area was generally lower in the skin samples (26) than in the carapace samples (51). Among carapace samples, the South African samples were the most taxa-abundant, whereas the lowest number of taxa characterized Florida and some of the Croatian samples (Fig 4A). Likewise, the number of genera differed among the populations (Fig 4B), being highest for the South African material. The carapace samples from Greece showed the highest diversity (median value 2,38) and evenness (0.61), while the samples from Florida had the lowest diversity (1,21) and evenness (0,32) (Fig 4C and 4D).

## Comparative analyses

The Sørenson similarity index showed that the Croatian and Greek samples were the most similar both at infrageneric (species level and below) and genus level, 35% and 62%, respectively. The lowest similarity on infrageneric level (almost 16%) is noted between the Florida and South Africa samples and on genus level between Florida and Greece (42.5%; Table 3).

According to the SIMPER analysis, Croatian samples had the lowest within-site similarity (average similarity 21,1%), followed by South African and Greek samples (49,3% and 57,8%, respectively), whereas Florida samples were the most homogenous ones (60,4%; S3 Table).

In general, the most abundant taxa in each sample group were also the ones contributing the most to the within-group similarity such as *Poulinea* CRO sp.1 and sp.2 for Croatia, *Nitzschia* cf. *insconspicua* and *Nitzschia* CRO sp.2 for Greece, *Chelonicola* SA sp.1. for South

**Table 2. The number of diatom taxa in the most diverse genera in samples from different localities.**

| Diatom genera | Overall | Croatia | Greece | South Africa | Florida |
|---|---|---|---|---|---|
| *Mastogloia* | 42 | 11 | 12 | 12 | 26 |
| *Navicula* | 32 | 11 | 13 | 11 | 6 |
| *Amphora* | 30 | 4 | 9 | 12 | 7 |
| *Nitzschia* | 30 | 13 | 10 | 12 | 8 |
| *Cocconeis* | 26 | 8 | 6 | 16 | 2 |
| *Licmophora* | 20 | 5 | 5 | 15 | 0 |
| *Diploneis* | 20 | 10 | 5 | 6 | 2 |
| *Seminavis* | 12 | 2 | 5 | 5 | 3 |
| *Achnanthes* | 10 | 4 | 2 | 9 | 1 |
| *Tryblionella* | 8 | 2 | 0 | 6 | 0 |
| other genera | 167 | 57 | 71 | 71 | 32 |

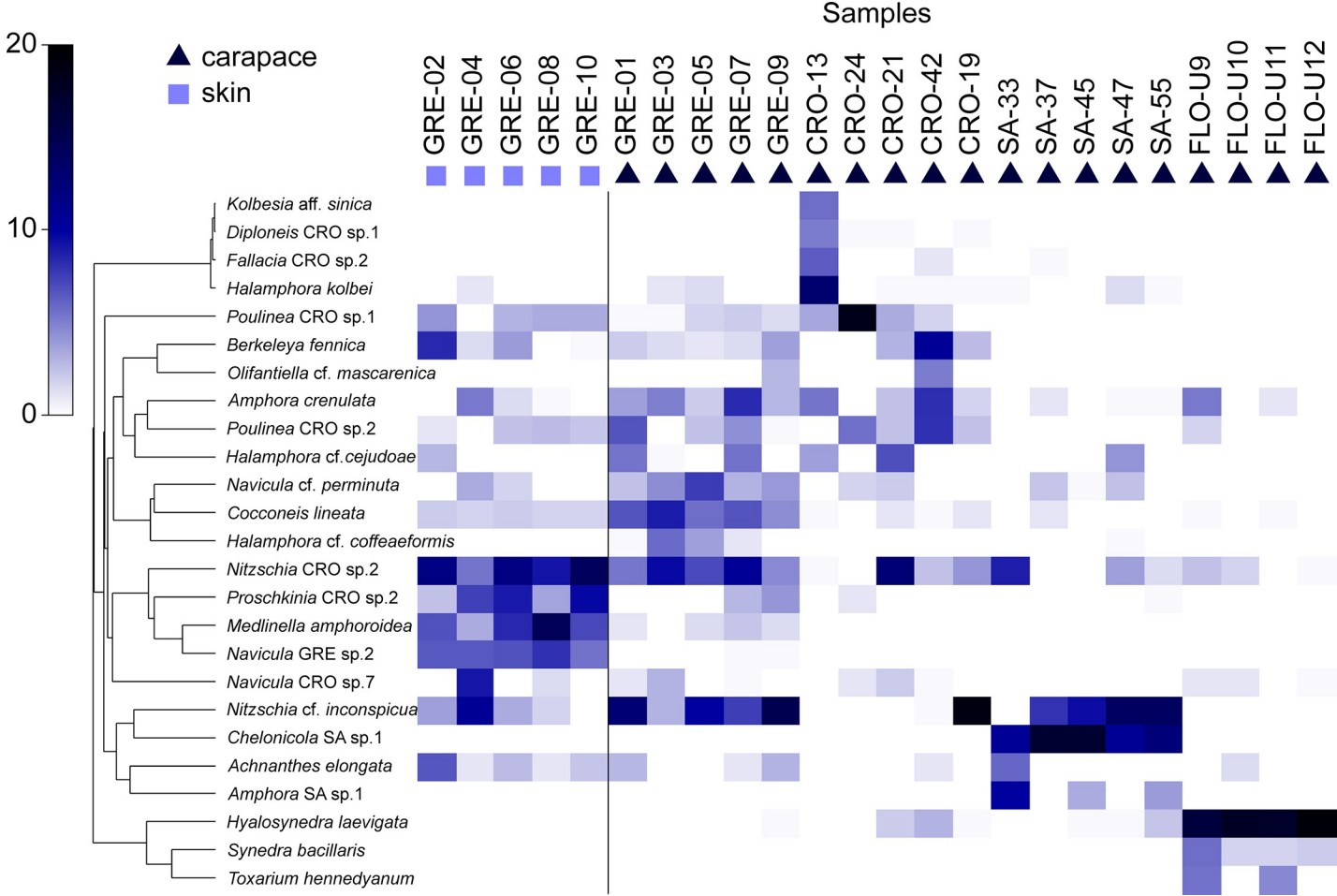

**Fig 3. The most abundant diatom taxa associated with loggerhead sea turtles.** Shade plot illustrating the 25 most abundant taxa recorded on loggerhead carapaces (triangle) and skins (square) from investigated localities based on square root-transformed abundance data. The white cells represent the absence of the taxa and the darkest cells the largest abundances. Taxa are ordered by a hierarchical cluster analysis of their mutual associations across samples based on Index of Association calculated on the standardized counts. CRO = Croatia, Adriatic Sea; FLO = Florida Bay, USA; GRE = Greece, Amvrakikos Gulf; SA = South Africa, Kosi Bay.

Africa and *Hyalosynedra laevigata* for Florida (S3 Table). SIMPER dissimilarity analysis (Table 4) showed that ten taxa contributed approx. 50% to the total differences observed between Greek and Croatian samples, with *Nitzschia* cf. in*conspicua* and *Cocconeis lineata* having the highest contributions. Samples from Florida differed from those from other locations mainly due to *Hyalosynedra laevigata* with 20.6%, 18.3%, and 21.7% contributions to the total dissimilarity observed between Florida and Croatia, Florida and Greece, and Florida and South Africa, respectively. South African samples were characterized by high abundances of *Chelonicola* SA sp. 1 that contributed 15.86%, 17,34%, and 15.67% to the total dissimilarity between South Africa and Croatia, South Africa and Florida, and South Africa and Greece. *Medlinella amphoroides*, *Nitzschia* cf. *inconspicua*, *Navicula* GRE sp. 2, and *Proschkinia* CRO sp. 2 were responsible for most of the differences between the skin and carapace diatom communities from Greece (Table 4).

Non-metric multidimensional scaling based on carapace diatom abundance data separated samples into five distinct groups (Fig 5A). The Florida cluster was the most distant from all remaining groups, while the South African, Greek, and Croatian samples were placed closer to

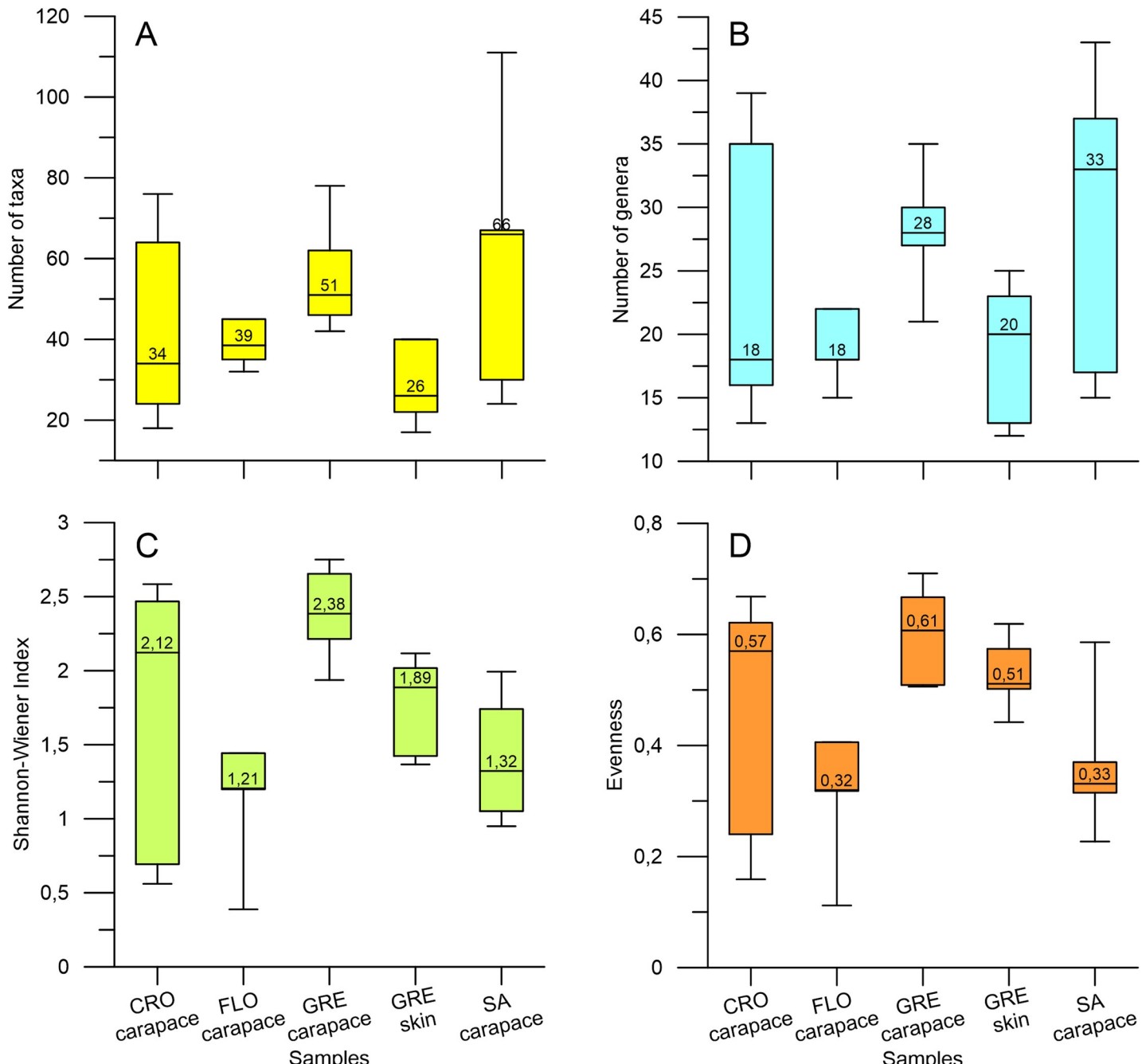

**Fig 4. Box and whisker plots of diatom community diversity across localities.** The diatom community diversity for loggerhead carapace samples from every locality, and the skin samples from Greece. (A) The number of taxa. (B) genera, (C) the Shannon-Wiener diversity index and D) evenness. Whiskers indicate maximum and minimum, the median value is denoted within the box. CRO = Croatia, Adriatic Sea; FLO = Florida Bay, USA; GRE = Greece, Amvrakikos Gulf; SA = South Africa, Kosi Bay.

each other, but in general, maintaining a good separation among the different localities. Croatian samples showed the lowest group homogeneity, with the main group of three samples and a single group comprised of one sample(CRO-13) and one sample placed in a different cluster (CRO 19). An additional nMDS analysis performed only on the Greek carapace and skin samples showed good separation of the two groups (Fig 5B).

Table 3. Sørensen-similarity index of the carapace samples between the different localities.

| Taxon level | Croatia | Florida | Greece |
|---|---|---|---|
| Florida | 20.37 | | |
| Greece | 35.02 | 21.20 | |
| South Africa | 25.57 | 15.85 | 22.22 |
| **Genus level** | **Croatia** | **Florida** | **Greece** |
| Florida | 46.91 | | |
| Greece | 62.14 | 42.50 | |
| South Africa | 54.05 | 45.45 | 49.09 |

The index was calculated at both species and genus level, expressed as percentages.

The PERMANOVA pair-wise test confirmed the significant effect of both the location (p<0,01) and the sea turtle body part (p = 0.008) on the associated diatoms (Table 5).

## Discussion

Loggerheads from the analyzed populations harbour a very diverse diatom flora with almost 400 taxa belonging to more than 100 genera. This number is most likely an underestimation of the exact taxon richness as a sampling of a limited number of turtle individuals may limit the number of diatom taxa found. Additionally, several taxa mostly belonging to the genera *Amphora*, *Navicula* and *Nitzschia* were grouped under a common name and detailed SEM and molecular analysis would be necessary to clarify their correct taxonomic identity. That would most likely result in the increase of the true taxon diversity. A clear example is *Nitzschia* cf. *inconspicua*, most likely representing a group of taxa difficult to disentangle rather than one single species. In the past, the *N. frustulum-inconspicua* group has been the subject of several taxonomic and molecular revisions resulting in the description of several new species and a better characterization of others such as *N. frustulum* (Kütz.) Grunow [40, 41] based on small morphological differences. *Nitzschia* cf. *inconspicua* was found in other epibiont diatom communities, for instance living predominantly on olive ridley turtles in Costa Rica [4]. Rivera et al. [23] applied both molecular and microscopic analyses of carapace samples from green turtles in the Marine Nature Park of Mayotte (Indian Ocean) and found *N. inconspicua* to be one of the most abundant taxa observed with a homogenous morphology across all seven investigated sea turtles. DNA analysis, on the contrary, indicated the presence of tens of OTU's (Operational Taxonomical Units), resulting in four groups implying a high (pseudo) cryptic diversity in *N. inconspicua*.

The observed taxon richness is clearly higher than currently observed from any other sea turtle species sampled so far. Majewska et al. [4] recorded only 21 taxa in 38 carapace samples from olive ridley sea turtles in Costa Rica whereas, in another study, Majewska et al. [16] reported 26 taxa belonging to 20 genera in 76 carapace samples from green turtles in Costa Rica and Iran. It is possible that the applied sampling technique in the latter two studies (i.e. use of razor blade or scalpel on a limited surface of the carapace) in contrast with the application of a toothbrush brushing off a larger surface influenced the observed taxon richness. Rivera et al. [23] used the toothbrush method to sample seven juvenile green turtles from Mayotte and observed 57 taxa. Our results also indicate a certain influence of the sampling technique. The Florida samples, collected with a cotton-tipped applicator, were the least diverse of all carapace samples. This method may have been too gentle to remove firmly attached, adnate diatom taxa, such as taxa from the genera *Cocconeis* and *Amphora* from the carapace, compared to the toothbrush and/or curette methods applied to sample the other

**Table 4. Contribution of species to dissimilarities between epizoic diatom assemblages of loggerhead populations–discriminating species.**

| Species | Average Abundance | Average Abundance2 | Average Dissimilarity | Dissimilarity/SD | Contribution % | Cumulative contribution % |
|---|---|---|---|---|---|---|
| **Croatia & Florida** | | | | | | |
| | **CRO** | **FLO** | | | | |
| *Hyalosynedra laevigata* | 1.04 | 17.56 | 19.05 | 3.41 | 20.61 | 20.61 |
| *Poulinea* CRO sp. 1 | 5.37 | 0.00 | 6.61 | 0.70 | 7.15 | 27.76 |
| *Nitzschia* cf. *inconspicua* | 3.83 | 0.00 | 5.29 | 0.49 | 5.72 | 33.48 |
| *Poulinea* CRO sp. 2 | 3.71 | 0.43 | 3.88 | 1.28 | 4.20 | 37.67 |
| *Nitzschia* CRO sp.2 | 3.89 | 1.10 | 3.78 | 0.92 | 4.09 | 41.76 |
| *Amphora crenulata* | 3.54 | 1.55 | 3.43 | 1.30 | 3.71 | 45.47 |
| *Berkeleya fennica* | 3.24 | 0.00 | 3.29 | 0.87 | 3.56 | 49.02 |
| *Synedra bacillaris* | 0.00 | 2.78 | 3.05 | 1.77 | 3.30 | 52.33 |
| **Croatia & Greece** | | | | | | |
| | **CRO** | **FLO** | | | | |
| *Nitzschia* cf. *inconspicua* | 3.83 | 9.61 | 8.05 | 2.13 | 10.82 | 10.82 |
| *Cocconeis lineata* | 0.49 | 6.45 | 5.00 | 3.65 | 6.72 | 17.54 |
| *Nitzschia* CRO sp.2 | 3.89 | 7.38 | 4.68 | 1.91 | 6.29 | 23.83 |
| *Poulinea* CRO sp. 1 | 5.37 | 1.12 | 4.33 | 0.69 | 5.81 | 29.64 |
| *Navicula* cf. *pavillardii* | 0.00 | 4.29 | 3.62 | 4.65 | 4.87 | 34.51 |
| *Navicula* cf. *perminuta* | 0.75 | 4.29 | 2.97 | 1.73 | 3.99 | 38.49 |
| *Poulinea* CRO sp. 2 | 3.71 | 2.73 | 2.59 | 1.37 | 3.49 | 41.98 |
| *Amphora crenulata* | 3.54 | 4.36 | 2.57 | 1.31 | 3.45 | 45.43 |
| *Berkeleya fennica* | 3.24 | 1.91 | 2.38 | 1.10 | 3.20 | 48.62 |
| *Halamphora kolbei* | 2.74 | 0.48 | 2.37 | 0.59 | 3.19 | 51.82 |
| **Florida & Greece** | | | | | | |
| | **FLO** | **GRE** | | | | |
| *Hyalosynedra laevigata* | 17.56 | 0.04 | 17.08 | 6.51 | 18.26 | 18.26 |
| *Nitzschia* cf. *inconspicua* | 0.00 | 9.61 | 9.44 | 2.13 | 10.09 | 28.34 |
| *Cocconeis lineata* | 0.11 | 6.45 | 6.13 | 4.28 | 6.55 | 34.89 |
| *Nitzschia* CRO sp.2 | 1.10 | 7.38 | 6.09 | 2.39 | 6.50 | 41.40 |
| *Navicula* cf. *pavillardii* | 0.00 | 4.29 | 4.17 | 5.22 | 4.45 | 45.85 |
| *Navicula* cf. *perminuta* | 0.00 | 4.29 | 4.13 | 2.43 | 4.41 | 50.26 |
| **Croatia & South Africa** | | | | | | |
| | **CRO** | **SA** | | | | |
| *Chelonicola* SA sp. 1 | 0.00 | 13.46 | 13.97 | 2.96 | 15.86 | 15.86 |
| *Nitzschia* cf. *inconspicua* | 3.83 | 9.05 | 9.49 | 1.76 | 10.77 | 26.64 |
| *Poulinea* CRO sp. 1 | 5.37 | 0.00 | 5.91 | 0.72 | 6.71 | 33.35 |
| *Nitzschia* CRO sp.2 | 3.89 | 2.77 | 4.07 | 1.12 | 4.62 | 37.97 |
| *Poulinea* CRO sp. 2 | 3.71 | 0.00 | 3.74 | 1.35 | 4.25 | 42.22 |
| *Amphora* SA sp. 1 | 0.00 | 3.44 | 3.38 | 0.97 | 3.84 | 46.05 |
| *Amphora crenulata* | 3.54 | 0.29 | 3.13 | 1.28 | 3.56 | 49.61 |
| *Berkeleya fennica* | 3.24 | 0.00 | 2.99 | 0.87 | 3.40 | 53.01 |
| **Florida & South Africa** | | | | | | |
| | **FLO** | **SA** | | | | |
| *Hyalosynedra laevigata* | 17.56 | 0.54 | 20.77 | 4.90 | 21.66 | 21.66 |
| *Chelonicola* SA sp. 1 | 0.00 | 13.46 | 16.62 | 3.08 | 17.34 | 39.00 |
| *Nitzschia* cf. *incospicua* | 0.00 | 9.05 | 11.17 | 1.76 | 11.65 | 50.65 |
| **Greece & South Africa** | | | | | | |
| | **GRE** | **SA** | | | | |

*(Continued)*

**Table 4.** (Continued)

| Species | Average Abundance | Average Abundance2 | Average Dissimilarity | Dissimilarity/SD | Contribution % | Cumulative contribution % |
|---|---|---|---|---|---|---|
| *Chelonicola* SA sp. 1 | 0.00 | 13.46 | 12.12 | 3.52 | 15.67 | 15.67 |
| *Cocconeis lineata* | 6.45 | 0.20 | 5.52 | 4.28 | 7.14 | 22.81 |
| *Nitzschia* CRO sp.2 | 7.38 | 2.77 | 4.84 | 1.72 | 6.26 | 29.07 |
| *Nitzschia* cf. *inconspicua* | 9.61 | 9.05 | 4.56 | 1.36 | 5.89 | 34.96 |
| *Navicula* cf. *pavillardii* | 4.29 | 0.04 | 3.78 | 5.30 | 4.88 | 39.85 |
| *Amphora crenulata* | 4.36 | 0.29 | 3.61 | 1.81 | 4.66 | 44.51 |
| *Amphora* SA sp. 1 | 0.00 | 3.44 | 2.95 | 0.98 | 3.82 | 48.33 |
| *Navicula* cf. *perminuta* | 4.29 | 0.98 | 2.92 | 1.57 | 3.77 | 52.10 |
| **Greece carapace & skin** | | | | | | |
| | **Carapace** | **Skin** | | | | |
| *Medlinella amphoroides* | 1.21 | 8.00 | 5.70 | 1.67 | 9.22 | 9.22 |
| *Nitzschia* cf. *inconspicua* | 9.61 | 3.90 | 5.70 | 1.48 | 9.21 | 18.43 |
| *Navicula* GRE sp.2 | 0.09 | 6.64 | 5.40 | 6.46 | 8.73 | 27.16 |
| *Proschkinia* CRO sp.2 | 1.39 | 6.42 | 4.31 | 1.65 | 6.97 | 34.13 |
| *Cocconeis lineata* | 6.45 | 1.84 | 3.78 | 3.21 | 6.11 | 40.25 |
| *Nitzschia* CRO sp.2 | 7.38 | 10.46 | 3.43 | 1.44 | 5.55 | 45.79 |
| *Amphora crenulata* | 4.36 | 1.37 | 2.97 | 1.55 | 4.81 | 50.60 |

Summary of SIMPER analysis of carapace and skin data based on Bray-Curtis dissimilarity, 70% cut off, taxa cumulatively contributing to the dissimilarity over 50% are shown. Croatia (CRO), Greece (GRE), South Africa (SA), Florida (FLO).

populations. The dominating genera in the Florida samples, *Hyalosynedra*, *Synedra* and *Toxarium*, are all large, erect diatom genera [3, 33], only attached by their apices to the surface and therefore more easily removed when using a cotton-tipped applicator. Brushing the surface with a hard toothbrush removes more efficiently the well-attached, adnate diatom taxa from the hard carapaces. Recently this method was designated as the standard sampling method for epizoic diatom communities [42].

Despite the high taxon richness, the percentage of the presumably truly epizoic taxa is rather low, although, we cannot be certain of an exact number of taxa that belong to that group. Several taxa were recently described from loggerhead samples from this dataset such as *Catenula exigua* K.Robert et al., *Planothidium kaetherobertianum* Van de Vijver & Bosak, and *Lucanicum ashworthianum* Majewska et al [21, 43, 44]. These taxa have not yet been found in epizoic samples from other localities and substrata. Similarly, the newly described *Proschkinia* species such as *P. vergostriata* Frankovich et al. and *P. sulcata* Majewska et al. have so far only been found on turtle carapaces and skin [19]. Thus, more sampling and analyses of marine benthic diatom communities from both biotic (including marine animals) and abiotic substrata will be necessary to determine the exact habitat preferences of these diatoms.

For turtles sampled in Greece, we sampled both skin and carapace. Interestingly, a large difference in the relative abundance of the presumably strictly epizoic taxa was observed. Skin communities were dominated by *Medlinella amphoroidea*, *Poulinea* spp, and *Achnanthes elongata*, all currently known only from sea turtles [12, 16, 21] whereas taxa belonging to common epiphytic and epipelic diatom genera, such as *Amphora*, *Halamphora*, *Berkeleya*, and *Cocconeis*, were more abundant in carapace samples. Skin sample GRE-04 and the matching carapace sample GRE-03 were collected from the same turtle. The high abundances of *Nitzschia* cf. *inconspicua* and *Navicula* sp.7 (Fig 3) present in the above-mentioned skin sample resulted in its grouping with carapace samples.

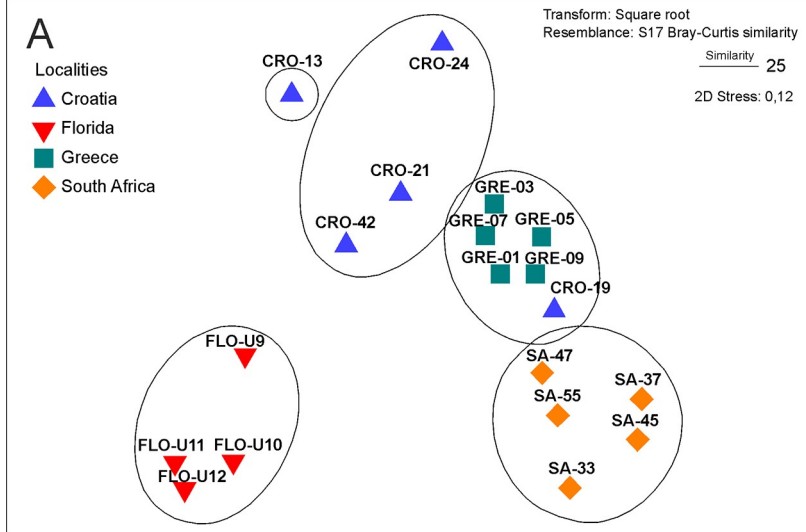

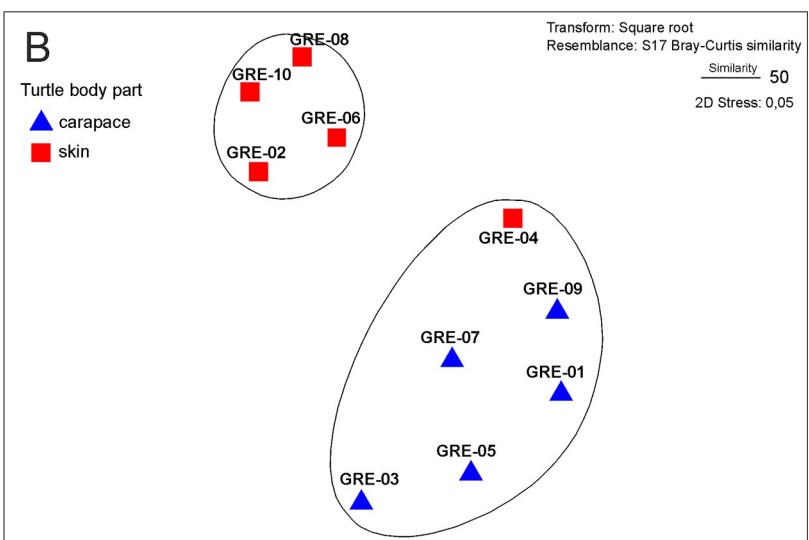

**Fig 5. Non-metric Multi-Dimensional Scaling (nMDS) plots of diatom assemblages on loggerhead turtles.** (A) Carapace samples from four localities. (B) Skin and carapace samples from Greece. The overlayed cluster analysis indicates grouping based on sample similarity of 25 and 50 in (A) and (B), respectively. CRO = Croatia, Adriatic Sea; FLO = Florida Bay, USA; GRE = Greece, Amvrakikos Gulf; SA = South Africa, Kosi Bay.

Many of the observed diatom taxa are probably 'ecological hitchhikers' using the animal surface as yet another hard substratum suitable for their development [45–48]. On the other hand, some species in common benthic genera may as well be the obligately epizoic taxa. This seems to be true for the two *Achnanthes* species that are regularly found in high abundances on various sea turtles from different oceans [16, 17, 20, pers.observations]. Moreover, several *Proschkinia* and *Craspedostauros* species described from the sea turtle carapaces and skin occur frequently on the animal substratum and, so far, have never been recorded from a non-animal habitat [19, 20, pers. observations].

The current results indicate that sea turtle skin is likely a much more specific substratum for diatom growth than the carapace, the latter sharing more similarities with other biotic (e.g. shells of snails and molluscs, barnacles) or abiotic surfaces (e.g. rocks). Strictly epizoic diatom

**Table 5. PERMANOVA analyses based on pairwise tests on square-root transformed data Sums of squares type: Type III (partial), fixed effects sum to zero for mixed terms.**

| Groups | df | t | P(perm) | Unique perms |
|---|---|---|---|---|
| Croatia, Florida | 7 | 2.3624 | 0.0078 | 126 |
| Croatia, Greece | 8 | 1.7282 | 0.0078 | 125 |
| Croatia, South Africa | 8 | 2.1666 | 0.0105 | 126 |
| Florida, Greece | 7 | 4.3461 | 0.0066 | 126 |
| Florida, South Africa | 7 | 3.8197 | 0.0083 | 126 |
| Greece, South Africa | 8 | 3.0402 | 0.0095 | 126 |
| Greece carapace & skin | 8 | 2.7412 | 0.008 | 126 |
| | **Average similarity between/within groups** | | | |
| | **Croatia** | **Florida** | **Greece** | **South Africa** |
| **Croatia** | 21.046 | | | |
| **Florida** | 7.5637 | 60.355 | | |
| **Greece** | 25.578 | 6.4227 | 57.828 | |
| **South Africa** | 11.931 | 4.1172 | 22.668 | 49.322 |
| | **carapace** | **skin** | | |
| **carapace** | 57.828 | | | |
| **skin** | 38.183 | 62.108 | | |

taxa develop well on the physiologically active substratum, whereas opportunistic benthic species, lacking some vital adaptations, may attach to the skin only temporarily when the external conditions are favourable. One of the striking examples is *Medlinella amphoroidea*, described from the skin of loggerheads in Florida [12]. In the present study, *Medlinella* was almost exclusively found in several skin samples but almost entirely absent from the carapace samples (only 10 valves found on the carapaces). Numerous opportunistic diatom taxa may end up on the carapaces of the loggerheads due to the foraging behavior of this turtle species [49, 50]. Other sea turtle species such as olive ridley and green turtle show a different feeding behavior and have a different diet [25], which may influence the epizoic diatom species composition. Robinson et al. [51] observed that the macro-epibiont diversity of nesting sea turtles is partially linked to the diversity of their foraging habitats. Thus, sea turtle species with more diverse foraging areas should have more diverse epibiont communities. Fuller et al. [52] reported that loggerheads host more macro-epibiotic species, such as barnacles, than green turtles. The authors of this study also suggest that the differences in epibiont communities between the two sea turtle species could be attributed to the difference in feeding behavior and diet, as adult loggerheads are benthic foragers, feeding by infaunal mining [53] and green turtles are herbivores, grazing on seagrass with little sediment disturbance [54]. Loggerheads often develop a rich macro-algal flora composed of a large number of filamentous algal taxa such as *Polysiphonia carettia* Hollenberg [55] or *Ectocarpus fasciculatus* Harvey. Epiphytic diatoms on these macroalgae, such as various species of *Cocconeis* or *Amphora*, although not directly attached to the animal body, may therefore further enrich the sea turtle-associated diatom community composition. As biofilm accumulates, the available and uncolonized substratum surface on the carapace decreases and so there will be also a decline in the relative abundance of strictly epizoic diatom taxa [16].

Thus, the behavior of the turtles and its impact on the attached diatom flora may explain why clear bioregionalism was found in the present study. Loggerhead samples from the Mediterranean localities (i.e. neighbouring Adriatic Sea and Amvrakikos Gulf), were found to be the most similar and distinct from both Southwest Indian (South Africa) and Northwest

Atlantic (Florida) samples. Amvrakikos Gulf (Greek samples) is connected with the Adriatic Sea (Croatian samples) via the Ionian Sea. Satellite tracking has revealed that loggerhead turtles in Amvrakikos Gulf generally remain resident in this area but do occasionally venture to the northern Adriatic to forage l [56]. Sample CRO-13 differed significantly from other Croatian diatom communities, as indicated by the nMDS plot. The sample was taken eight months after the injured turtle arrived in the rescue center and after it was cleaned from its original epizoic biofilm due to standard procedures applied at the facility. The observed diatom flora showed a remarkable similarity to the diatom flora that was growing on the walls of the plastic housing tank in which the turtle was undergoing rehabilitation for several months (Bosak, pers. observation). This may reflect a rather easy transfer of diatom taxa present on the objects within the enclosure to the carapace surface of captive turtles, especially if the new environment restrains the animal from exhibiting its natural behaviour (e.g. feeding by diving, fast-swimming). As already proposed by Holmes et al. [8], and later by Wetzel et al. [11] and Majewska et al. [16], transfer of surface-associated diatoms between different animals occurs likely through body-to-body contact. It is plausible that physical contact will also be required for a diatom transfer between the animal host and inanimate objects.

A considerable part of the observed diatom taxa in the samples from Croatia and Greece was illustrated previously in the monograph of Álvarez-Blanco & Blanco [34] on the benthic diatom flora of the Mediterranean coasts. On the other hand, the dominating genera in the samples from Florida include *Hyalosynedra*, *Synedra*, *Toxarium* and *Mastogloia*, the latter present with a fairly large number of species, are often reported from the Florida Bay region [33, 57]. These observations seem to support the previously suggested hypothesis [14] that diatom composition may serve as a biogeographical indicator of the whereabouts of sea turtles, especially loggerheads that host particularly diverse diatom communities. By comparing the diatom flora on the sea turtle with known marine benthic diatom floras worldwide, it may be possible to detect where the loggerhead has been residing. Studies on epiphytic diatoms show that the epibiotic diatom communities may vary greatly depending on geographical locality and external environmental conditions [4, 16]. A follow-up study should explore both the epibiotic loggerhead flora and the local benthic (including diverse abiotic substrata and hard-surfaced animals) diatom communities. Additionally, a study combining the analysis of the epibiotic diatom flora and satellite tracking may be an interesting research venue.

## Conclusion

The diatom flora on the carapaces and skin of loggerhead sea turtles from geographically distinct locations shows a remarkable diversity and a generally low similarity. Loggerheads from the same location share a common pool of diatoms, showing clear bioregionalism, and diatom communities on sea turtles from more distant regions show less similarity between each other than those from neighbouring areas.

In many cases, the presumably truly epizoic species were outnumbered by the local benthic taxa and had only a minor contribution to the sea turtle-associated diatom floras. This may be partially explained by the frequent physical contact with a variety of substrata occurring during the specific foraging activities of loggerheads. Although species-rich diatom communities are found on both the sea turtle carapace and skin, those associated with the latter appear to be less diverse with a higher abundance of the presumably exclusively epizoic taxa.

Loggerheads serve as reservoirs and probable vectors for diverse and often unique diatom communities. This ecological role of sea turtles is still poorly understood and rarely discussed, and future studies are required to throw more light on the sea turtle contribution to the benthic diatom dispersal and their modern biogeography.

## Supporting information

**S1 Table. List of taxonomic publications used for identification of diatom taxa on loggerhead sea turtles.**
(PDF)

**S2 Table. List of 397 taxa observed in the carapace and skin samples of the four sampling localities.** Presumably exclusively epizoic taxa are indicated in bold.
(PDF)

**S3 Table. Similarity analysis of loggerhead epizoic diatom assemblages within each sampling locality–typical species.** SIMPER analysis was based on Bray-Curtis similarity, 70% cut off, taxa cumulatively contributing to the similarity over 70% are shown. Croatia (CRO), Greece (GRE), South Africa (SA), Florida (FLO).
(PDF)

## Acknowledgments

Ronel Nel and Diane Z. M. Le Gouvello du Timat (Nelson Mandela University, South Africa) are thanked for their help during the material collection in South Africa and obtaining the necessary sampling permits. We thank Brian Stacy of the US National Marine Fisheries Service for the collection of diatom samples and Allen Foley of the Florida Fish and Wildlife Conservation Commission and Jennifer Keene of the University Of Florida College of Veterinary Medicine for allowing us to receive samples from captured loggerhead turtles during the annual Florida Bay sea turtle survey. For the Croatian samples, we are thankful to Milena Mičić and Karin Gobić Medica as well the rest of the staff from Marine Turtle Rescue Centre, Aquarium Pula. ARCHELON volunteers are thanked for their help during Amvrakikos Gulf sampling activities. Mrs Myriam de Haan is thanked for preparing the samples for LM and SEM analysis.This is a contribution 188 from the Division of Coastlines and Oceans of the Institute of Environment at Florida International University.

## Author Contributions

**Conceptualization:** Bart Van de Vijver, Roksana Majewska, Sunčica Bosak.

**Data curation:** Käthe Robert, Roksana Majewska, Thomas A. Frankovich.

**Formal analysis:** Bart Van de Vijver, Käthe Robert, Sunčica Bosak.

**Funding acquisition:** Bart Van de Vijver, Sunčica Bosak.

**Investigation:** Bart Van de Vijver, Käthe Robert, Sunčica Bosak.

**Methodology:** Bart Van de Vijver, Käthe Robert, Roksana Majewska, Thomas A. Frankovich, Sunčica Bosak.

**Project administration:** Bart Van de Vijver, Sunčica Bosak.

**Resources:** Bart Van de Vijver, Roksana Majewska, Thomas A. Frankovich, Aliki Panagopoulou, Sunčica Bosak.

**Supervision:** Bart Van de Vijver, Roksana Majewska, Thomas A. Frankovich, Sunčica Bosak.

**Validation:** Aliki Panagopoulou, Sunčica Bosak.

**Visualization:** Käthe Robert, Sunčica Bosak.

**Writing – original draft:** Bart Van de Vijver, Käthe Robert, Roksana Majewska, Thomas A. Frankovich, Sunčica Bosak.

**Writing – review & editing:** Bart Van de Vijver, Roksana Majewska, Thomas A. Frankovich, Aliki Panagopoulou, Sunčica Bosak.

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
