## [Decision Letter · Decision Letter 0]

16 Apr 2020

PONE-D-20-05852

Diversity of diatom communities associated with loggerhead sea turtles (Carreta carreta)

PLOS ONE

Dear Dr. Bosak,

Thank you for submitting your manuscript to PLOS ONE. After careful consideration, we feel that it has merit but does not fully meet PLOS ONE’s publication criteria as it currently stands. Therefore, we invite you to submit a revised version of the manuscript that addresses the points raised during the review process.

We would appreciate receiving your revised manuscript by May 31 2020 11:59PM. To enhance the reproducibility of your results, we recommend that if applicable you deposit your laboratory protocols in protocols.io, where a protocol can be assigned its own identifier (DOI) such that it can be cited independently in the future. For instructions see: http://journals.plos.org/plosone/s/submission-guidelines#loc-laboratory-protocols

We look forward to receiving your revised manuscript.

Kind regards,

Vona Méléder, Ph.D.

Academic Editor

PLOS ONE

2. Please include a copy of Table 5 which you refer to in your text on page 17.

3. We note that Figure 1 in your submission contain map images which may be copyrighted. All PLOS content is published under the Creative Commons Attribution License (CC BY 4.0), which means that the manuscript, images, and Supporting Information files will be freely available online, and any third party is permitted to access, download, copy, distribute, and use these materials in any way, even commercially, with proper attribution. For these reasons, we cannot publish previously copyrighted maps or satellite images created using proprietary data, such as Google software (Google Maps, Street View, and Earth). For more information, see our copyright guidelines: http://journals.plos.org/plosone/s/licenses-and-copyright.

Reviewers' comments:

Reviewer's Responses to Questions

**Comments to the Author**

1. Is the manuscript technically sound, and do the data support the conclusions?

Reviewer #1: Yes

Reviewer #2: Yes

2. Has the statistical analysis been performed appropriately and rigorously? 

Reviewer #1: Yes

Reviewer #2: Yes

3. Have the authors made all data underlying the findings in their manuscript fully available?

Reviewer #1: Yes

Reviewer #2: Yes

4. Is the manuscript presented in an intelligible fashion and written in standard English?

Reviewer #1: Yes

Reviewer #2: Yes

5. Review Comments to the Author

Reviewer #1: Review of:

Diversity of diatom communities associated with loggerhead sea turtles (Carreta carreta)

Van de Vijver et al. present a well-written, interesting, and valuable study. My only major comment is that I believe that at the end of the discussion, I think the paper would benefit from having some more discussion about how the findings of this study could applied. The authors do a great job of explaining the why they think that diatom community structures differ between turtles from different regions; however, they do not really expand on the bigger picture implications of this. I think a rather simple fix would be to acknowledge that as diatom communities may reflect foraging patterns of these turtles, then diatom communities could be used as behavioral indicators for these species.

All other comments are relatively minor and are listed below:

Line 25: “Epizoic diatoms form an important part of micro-epibiota on marine vertebrates” should be “Epizoic diatoms form an important part of the micro-epibiota of marine vertebrates”.

Line 29 - 30: “Almost 400 diatom taxa belonging to more than 100 genera have been observed.” Because this sentence is written in the passive tense, it is not clear whether you are referring to only the diatom taxa observed in this study or all the taxa reported in all the published literature on this subject. I would switch this sentence to the active tense. I.e. “ We observed almost 400 diatom taxa belonging to more than 100 genera.” There are also several other examples in the document where it would be clearer to use the active tense instead of the passive tense. Please change throughout.

Line 31: “Diatom communities from Greece and Croatia showed the highest similarity, clearly differing from the communities observed in the samples from South Africa and Florida.” should be “Diatom communities from Greece and Croatia showed the highest similarity and were statistically different to those recorded from South Africa and Florida.”

Line 32: “Part of the difference in diversity and composition may be attributed to different sampling techniques.” I think this could be rephrased as, at present, it largely undermined the entire study. How about “While sampling technique different between locations, and this may affect some of various we observed between localities, we believe that geography played a larger role”.

Line 38: The abstract would strongly benefit from a concluding sentence that explains what are the implications of your findings.

Line 46: “everywhere where” should be “wherever”.

Line 416: “Has” should be “have”.

Line 429: The authors appear to have cited the wrong study here. The authors state “Robinson et al. [14] observed that the macro-epibiont diversity of nesting sea turtles is partially linked to the diversity of their foraging habitats.” However, they cite another study by this author that looks at diatom and noi macro-epibiont diversity. I believe the authors were aiming to cite this study instead “Robinson NJ, Lazo-Wasem EA, Paladino FV, Zardus JD, Pinou T (2017) Assortative epibiosis of leatherback, olive ridley and green sea turtles in the Eastern Tropical Pacific. Journal of the Marine Biological Association of the United Kingdom 97(6):1233-40.”.

Reviewer #2: The manuscript is very well written and the graphs are all very clear. Great effort was taken to identify the taxa and the number of taxa found is quite impressive! The abstract provides a nice description of the manuscript.

The title could be a bit more descriptive – something like ‘Geographical variation in the diverse diatom communities associated …’.

The goals of the study are simply are simply to document the diatom communities on the turtles and see whether different areas of the turtles have different diatom composition. The manuscript would be greatly improved with a hypothesis to test. For example, although a previous study demonstrated that turtles of different species in the same area have similar diatom epibionts, but you might anticipate differences among turtles from different areas, if these turtles are truly geographically separated. Also, why might you expect skin and carapace areas to have different diatom populations?

In the Discussion, there seems to be an ‘all or none’ sort of definition to epizoic taxa -as if they should never be found on non-animal surfaces. Benthic and even attached diatoms often become ‘planktonic’ – at least until they settle on a surface. If epizoic species thereby end up on non-animal sources and survive, this may not mean that they are not epizoic. Especially as the processing method does not allow distinguishing between empty frustules and live diatoms that are not thriving. Similarly, I would expect many benthic diatoms to do well on the carapace (which is relatively inert). The skin is more interesting, as I suspect there is more sloughing and secretions from the turtle – and so it is not surprising that more presumed epizoic taxa were found on the skin samples.

Line 64. ‘accidently’ is probably the wrong word. Maybe ‘haphazardly’ might be a better choice.

Lines 64-65. I suspect that true planktonic and tychoplanktonic species may also settle on turtles (no contact with solid surfaces needed).

Line 97. Are the Adriatic Sea and Amvrakikos Gulf turtles from different subpopulations? (The locations are fairly close and the species is ‘highly migratory’ (line 83). This is mentioned in the Discussion but should be added her.

Line 217. Delete the word ‘up’.

Table 2. In the legend, ‘Number of diatom genera’, should be ‘Number of diatom taxa in the most diverse genera’.

Lines 243-244. ‘most frequently occurring’ could mean either having the highest total count or occurring in the most samples or at the most sites. Suggestion: for the first % listed, include the units (percent of what?).

Lines 269-270. Maybe drop the color references, as they all look purple – and (I think) this blue vs purple refers only to the symbols in the headings and not the colors used to indicate abundance – and symbol shape is enough.

Line 272. ‘the fill cells the largest abundance’. The ‘largest abundance in the matrix’ can only be one cell – and all the cells are filled (whether by while or by color). Reword.

Line 254. Replace ‘the latter’ with ‘Florida samples’

Lines 281-282. This excludes the GRE skin samples, which had the lowest median value.

Line 305. ‘were the most diverse’ was a bit tricky to decipher, as it wasn’t obvious that the comparison was among the samples within each site. Maybe ‘had the lowest within-site similarity’.

Line 334. There are 5 groups, one of which is a single sample (I don’t think I would call it an outlier).

Also Line 337. ‘with a main group of 3 samples, a single group comprised of one sample, and…

Fig. 5B. If your skin samples were numbered as pairs with the carapace samples, the matching carapace sample to the skin sample within the carapace cluster (GRE-04) would be GRE-03 – which are the 2 farthest apart samples in the cluster. If this is correct, it would be worth noting (especially since the location of GRE-04 within the carapace cluster wasn’t noted).

Line 359. Sampling only 5 individuals per species also limits the number of species found (as indicated by the relatively low similarity among individuals within some of the sites).

Line 474. The diatom floras might also indicate segregation among populations of sea turtles (especially benthic feeding species).

6. PLOS authors have the option to publish the peer review history of their article (what does this mean?). If published, this will include your full peer review and any attached files.

Reviewer #1: Yes: Nathan Jack Robinson

Reviewer #2: Yes: Elizabeth Bergey

---

## [Author Response · Author response to Decision Letter 0]

13 May 2020

PONE-D-20-05852

Diversity of diatom communities associated with loggerhead sea turtles (Carreta carreta)

PLOS ONE

Response to reviewers

Dear Editor,

Thank you for the consideration of our manuscript for the publication in PLOS ONE. The authors thank the two reviewers for their thorough review, comments, and suggestions. We believe that the revisions substantially improved the manuscript. We made the requested changes, both in the text and in the figures, please find our answer (A) to the journal requirements (J) and reviewers’ suggestions (R) below. 

J: 1. Please ensure that your manuscript meets PLOS ONE's style requirements, including those for file naming. The PLOS ONE style templates can be found at http://www.plosone.org/attachments/PLOSOne_formatting_sample_main_body.pdf and http://www.plosone.org/attachments/PLOSOne_formatting_sample_title_authors_affiliations.pdf

A: Checked and changed accordingly

J: 2. Please include a copy of Table 5 which you refer to in your text on page 17.

A: The Table caption is corrected to 5. 

J: 3. We note that Figure 1 in your submission contain map images which may be copyrighted. All PLOS content is published under the Creative Commons Attribution License (CC BY 4.0), which means that the manuscript, images, and Supporting Information files will be freely available online, and any third party is permitted to access, download, copy, distribute, and use these materials in any way, even commercially, with proper attribution. For these reasons, we cannot publish previously copyrighted maps or satellite images created using proprietary data, such as Google software (Google Maps, Street View, and Earth). For more information, see our copyright guidelines: http://journals.plos.org/plosone/s/licenses-and-copyright. We require you to either (1) present written permission from the copyright holder to publish these figures specifically under the CC BY 4.0 license, or (2) remove the figures from your submission: 

A: The figure 1. has been replaced with the figure that complies with CC BY 4.0 license according to the recommendation from the Editors office. The caption now says:

“Fig 1. The sampling areas of loggerhead sea turtles. (A) Red dots indicate locations of sampled loggerheads. Inserts show details of the sampling locations: (B) Amvrakikos Gulf, Greece; (C) Adriatic Sea, Croatia; (D) Florida Bay, USA; (E) Kosi Bay, South Africa. The maps were made with Natural Earth. Free vector and raster map data @ naturalearthdata.com.”

Reviewer #1: 

R: Van de Vijver et al. present a well-written, interesting, and valuable study. My only major comment is that I believe that at the end of the discussion, I think the paper would benefit from having some more discussion about how the findings of this study could applied. The authors do a great job of explaining the why they think that diatom community structures differ between turtles from different regions; however, they do not really expand on the bigger picture implications of this. I think a rather simple fix would be to acknowledge that as diatom communities may reflect foraging patterns of these turtles, then diatom communities could be used as behavioral indicators for these species.

A: Although we appreciate the reviewer’s kind words and suggestions, we believe that if we include the conclusion proposed by the reviewer, we should have observed behavior and foraging patterns of the sea turtles that we investigated, and this was not done in our study. Also, this paper includes only one sea turtle species and we think more detailed research including comparisons between different host species with different habitats and behaviors is needed. The main objective of this study was to provide a baseline for the future epizoic diatom studies on loggerheads and we do not want to speculate about bigger implications until they can be proven. However, we have included the following sentences at the end of the abstract that we feel are appropriate conclusions drawn from the results of this study.

“Our results indicate that epizoic diatom communities differ according to host species geographical location and host substrate (skin vs. carapace). The relative abundances of common benthic diatoms and putative exclusive epizoic taxa may inform about host habitat use or behavior though detailed comparisons between different host sea turtle species have yet to be performed.”

R: All other comments are relatively minor and are listed below:

Line 25: “Epizoic diatoms form an important part of micro-epibiota on marine vertebrates” should be “Epizoic diatoms form an important part of the micro-epibiota of marine vertebrates”.

A: Corrected

R: Line 29 - 30: “Almost 400 diatom taxa belonging to more than 100 genera have been observed.” Because this sentence is written in the passive tense, it is not clear whether you are referring to only the diatom taxa observed in this study or all the taxa reported in all the published literature on this subject. I would switch this sentence to the active tense. I.e. “ We observed almost 400 diatom taxa belonging to more than 100 genera.” There are also several other examples in the document where it would be clearer to use the active tense instead of the passive tense. Please change throughout.

A: We have changed suggested use from passive voice to active, as the reviewer suggested. 

R: Line 31: “Diatom communities from Greece and Croatia showed the highest similarity, clearly differing from the communities observed in the samples from South Africa and Florida.” should be “Diatom communities from Greece and Croatia showed the highest similarity and were statistically different to those recorded from South Africa and Florida.”

A: Corrected

R: Line 32: “Part of the difference in diversity and composition may be attributed to different sampling techniques.” I think this could be rephrased as, at present, it largely undermined the entire study. How about “While sampling technique different between locations, and this may affect some of various we observed between localities, we believe that geography played a larger role”.

A: Thank you for this comment, we have changed this sentence according to your suggestion.

R: Line 38: The abstract would strongly benefit from a concluding sentence that explains what are the implications of your findings.

A: We added the following sentences: 

“Our results indicate that epizoic diatom communities differ according to host species geographical location and host substrate (skin vs. carapace). The relative abundances of common benthic diatoms and putative exclusive epizoic taxa may inform about host habitat use or behavior though detailed comparisons between different host sea turtle species have yet to be performed.”

R: Line 46: “everywhere where” should be “wherever”.

A: Corrected

R: Line 416: “Has” should be “have”.

A: Corrected

R: Line 429: The authors appear to have cited the wrong study here. The authors state “Robinson et al. [14] observed that the macro-epibiont diversity of nesting sea turtles is partially linked to the diversity of their foraging habitats.” However, they cite another study by this author that looks at diatom and noi macro-epibiont diversity. I believe the authors were aiming to cite this study instead “Robinson NJ, Lazo-Wasem EA, Paladino FV, Zardus JD, Pinou T (2017) Assortative epibiosis of leatherback, olive ridley and green sea turtles in the Eastern Tropical Pacific. Journal of the Marine Biological Association of the United Kingdom 97(6):1233-40.”.

A: Yes, thank you for this observation, we changed the citation accordingly. 

R: Reviewer #2: The manuscript is very well written and the graphs are all very clear. Great effort was taken to identify the taxa and the number of taxa found is quite impressive! The abstract provides a nice description of the manuscript.

A: Thank you.

R: The title could be a bit more descriptive – something like ‘Geographical variation in the diverse diatom communities associated …’.

A: The title is changed to:

“Geographical variation in the diatom communities associated with loggerhead sea turtles (Carreta carreta)”

R: The goals of the study are simply to document the diatom communities on the turtles and see whether different areas of the turtles have different diatom composition. The manuscript would be greatly improved with a hypothesis to test. For example, although a previous study demonstrated that turtles of different species in the same area have similar diatom epibionts, but you might anticipate differences among turtles from different areas, if these turtles are truly geographically separated. Also, why might you expect skin and carapace areas to have different diatom populations?

A: Thank you for your comment, however, as we have stated at the end of our introduction, the objective of the present study was to provide a baseline data for the studies of diatom diversity on loggerheads, therefore the hypothesis such as suggested could not been put forward. We expect exactly to do just that in our following manuscripts, investigate differences and links between different host species and their habitats. The different flora on the skin and carapace was expected due to the characteristics of the substratum as we have explained in the discussion, the skin is a physiologically more active substratum and favors the development of epizoic taxa. 

R: In the Discussion, there seems to be an ‘all or none’ sort of definition to epizoic taxa -as if they should never be found on non-animal surfaces. Benthic and even attached diatoms often become ‘planktonic’ – at least until they settle on a surface. If epizoic species thereby end up on non-animal sources and survive, this may not mean that they are not epizoic. Especially as the processing method does not allow distinguishing between empty frustules and live diatoms that are not thriving. Similarly, I would expect many benthic diatoms to do well on the carapace (which is relatively inert). The skin is more interesting, as I suspect there is more sloughing and secretions from the turtle – and so it is not surprising that more presumed epizoic taxa were found on the skin samples.

A: Yes, it is true, we cannot know for a fact that taxa we found as epizoic cannot survive on other non-animal surfaces, however, to our knowledge, they have not been found yet there, and apparently, these species are developing large populations on animal surfaces, therefore we label them as epizoic. 

R: Line 64. ‘accidently’ is probably the wrong word. Maybe ‘haphazardly’ might be a better choice.

A: Thank you, the word has been replaced

R: Lines 64-65. I suspect that true planktonic and tychoplanktonic species may also settle on turtles (no contact with solid surfaces needed). 

A: Yes, that is true, as the true planktonic species such as Chaetoceros and Pseudonitzschia can settle (resting stages, vegetative cells) on the marine sediment bottom, especially in the bloom period, they can also be found in samples scraped from sea turtles as they get caught on these animals (as a sediment trap) instead of travelling all the way to the bottom. However, in the sample list that we analyzed as a part of this study, we did not find any examples of these true planktonic taxa. 

R: Line 97. Are the Adriatic Sea and Amvrakikos Gulf turtles from different subpopulations? (The locations are fairly close and the species is ‘highly migratory’ (line 83). This is mentioned in the Discussion but should be added here.

A: Yes, we added this information as reviewer suggested.

R: Line 217. Delete the word ‘up’.

A: Deleted

R: Table 2. In the legend, ‘Number of diatom genera’, should be ‘Number of diatom taxa in the most diverse genera’.

A: Yes, corrected.

R: Lines 243-244. ‘most frequently occurring’ could mean either having the highest total count or occurring in the most samples or at the most sites. Suggestion: for the first % listed, include the units (percent of what?).

A: We have included: Nitzschia CRO sp.2 (present 83.3% of all samples)

R: Lines 269-270. Maybe drop the color references, as they all look purple – and (I think) this blue vs purple refers only to the symbols in the headings and not the colors used to indicate abundance – and symbol shape is enough. R: Line 272. ‘the fill cells the largest abundance’. The ‘largest abundance in the matrix’ can only be one cell – and all the cells are filled (whether by while or by color). Reword.

A: We have corrected the Figure caption following both comments:

“Fig. 3. The most abundant diatom taxa associated with loggerhead sea turtles 

Shade plot illustrating the 25 most abundant taxa recorded on loggerhead carapaces (triangle) and skins (square) from investigated localities based on square root-transformed abundance data. The white cells represent the absence of the taxa and the darkest cells the largest abundances.. Taxa are ordered by a hierarchical cluster analysis of their mutual associations across samples based on Index of Association calculated on the standardized counts. CRO = Croatia, Adriatic Sea; FLO = Florida Bay, USA; GRE = Greece, Amvrakikos Gulf; SA = South Africa, Kosi Bay”

R: Line 254. Replace ‘the latter’ with ‘Florida samples’

A: Yes, corrected.

R: Lines 281-282. This excludes the GRE skin samples, which had the lowest median value.

A: Yes, we added at the beginning of the sentence the clarification that it implies only the samples from the carapace.

R: Line 305. ‘were the most diverse’ was a bit tricky to decipher, as it wasn’t obvious that the comparison was among the samples within each site. Maybe ‘had the lowest within-site similarity’.

A: Thank you, corrected.

R: Line 334. There are 5 groups, one of which is a single sample (I don’t think I would call it an outlier).

Also Line 337. ‘with a main group of 3 samples, a single group comprised of one sample, and…

A: Thank you, corrected.

R: Fig. 5B. If your skin samples were numbered as pairs with the carapace samples, the matching carapace sample to the skin sample within the carapace cluster (GRE-04) would be GRE-03 – which are the 2 farthest apart samples in the cluster. If this is correct, it would be worth noting (especially since the location of GRE-04 within the carapace cluster wasn’t noted).

A: We noted the position of GRE-04 in the discussion and offered a possible explanation:

“Skin sample GRE-04 and the matching carapace sample GRE-03 were collected from the same turtle. The high abundances of Nitzschia cf. inconspicua and Navicula sp.7. (Fig 3.) present in above-mentioned skin sample resulted in its grouping with carapace samples.” 

R: Line 359. Sampling only 5 individuals per species also limits the number of species found (as indicated by the relatively low similarity among individuals within some of the sites).

A: Yes, we agree, the following sentence was added to the discussion. 

“This number is most likely an underestimation of the exact richness as sampling of a limited number of turtle individuals may limit the number of diatom taxa found.”

R: Line 474. The diatom floras might also indicate segregation among populations of sea turtles (especially benthic feeding species).

A: We agree as we wrote in our discussion: “Thus, the behavior of the turtles and its impact on the attached diatom flora may explain why a clear bioregionalism was found in the present study. “

---

## [Decision Letter · Decision Letter 1]

5 Jun 2020

PONE-D-20-05852R1

Geographical variation in the diatom communities associated with loggerhead sea turtles (Carreta carreta)

PLOS ONE

Dear Dr. Bosak,

Thank you for submitting your manuscript to PLOS ONE. After careful consideration, we feel that it has merit but does not fully meet PLOS ONE’s publication criteria as it currently stands. Therefore, we invite you to submit a revised version of the manuscript that addresses the points raised during the review process.

We look forward to receiving your revised manuscript.

Kind regards,

Vona Méléder, Ph.D.

Academic Editor

PLOS ONE

Additional Editor Comments (if provided):

Dear authors,

Several issues have been found throughout the document, certainly based on the fact that many of the revisions have created even more errors in the text instead of fixing them. Could you please take the comments from reviewer # 1 into account, then read your ms carefully before resubmitting.

Reviewers' comments:

Reviewer's Responses to Questions

**Comments to the Author**

1. If the authors have adequately addressed your comments raised in a previous round of review and you feel that this manuscript is now acceptable for publication, you may indicate that here to bypass the “Comments to the Author” section, enter your conflict of interest statement in the “Confidential to Editor” section, and submit your "Accept" recommendation.

Reviewer #1: (No Response)

Reviewer #2: All comments have been addressed

2. Is the manuscript technically sound, and do the data support the conclusions?

Reviewer #1: Yes

Reviewer #2: Yes

3. Has the statistical analysis been performed appropriately and rigorously? 

Reviewer #1: Yes

Reviewer #2: Yes

4. Have the authors made all data underlying the findings in their manuscript fully available?

Reviewer #1: Yes

Reviewer #2: Yes

5. Is the manuscript presented in an intelligible fashion and written in standard English?

Reviewer #1: No

Reviewer #2: Yes

6. Review Comments to the Author

Reviewer #1: Review Of

Geographical variation in the diatom communities associated with loggerhead sea turtles (Carreta carreta)

I still agree that the study presented by Van de Vijver et al. is interesting and worthy of publication. Nevertheless, on a detailed read through, I have noted several typos, errors, and situations were the language could be improved. Thus, I think that the text needs some considerable improvements to be at the level where it would be ready for publication. I have listed several comments below. On top of this, I would also recommend that the authors also take the time to give this manuscript a detailed read through before resubmission to make sure that the text is as clear and understandable as possible.

Title: “Carreta caretta” should be “caretta caretta”.

Line 25: I would delete this first sentence. The second sentence is a perfect location for this manuscript to start.

Line 28: “skins and shells” do not need to be pluralized in this context.

Line 28: Please include the scientific names after you first provide the species common name.

Line 33: “While sampling technique differed between locations, and this may affect some of variations we observed between localities, we believe that geography played a larger role” should be “We believe that part of this variation was due to differences in sampling techniques; however, we still believe that geography had an important role.”

Line 35 – 37: “Only 5 000 of the known diatom species are considered to be marine with 50 another 50 000 diatom species still to be discovered and described in the marine realm” should be “Of these, around 55,000 are estimated to exist in marine habitats. To date however, less than 5,000 of marine diatoms have been described.”

Line 51-52: This sentence should be moved to be the second sentence in this paragraph.

Line 62: You already mention that diatoms are found on whales and sea turtles in Line 57. Thus, it is a little confusing to again mention all the different species that diatoms are found on just a few sentences later. I would delete Lines 62 – 65 and start by talking about sea turtles.

Line 70: Delete “being”.

Line 82: This should be written in the past not present tense.

Line 111 – 112: I would argue that this is unnecessary and should be deleted.

Line 115: I do not think that originates is the appropriate word here. The population does not breed in Amvarikos, it just forages here.

Line 128: “All samples were collected in a non-invasive way for the animal.” I would delete this sentence as you explain this in the methods.

Line 150: “In total, 25 samples were analyzed, twenty from the carapace and five from the skin.” This is a very confusing way to write this. I would change it to “In total, we collected 25 samples from loggerhead skin. In addition, for five of these animals we were also able to simultaneously sample their carapaces for diatoms.”

Table 1: This could be condensed as much information is repeated. How about one line for each sampling location with ranges?

Line 159-160: “Samples were processed following the methods described by Hasle and Syvertsen [30] for South African samples and van der Werff [31].” This is a confusing way to put this. Are you trying to say that you used the Hasle and Syvertsen method only for South African samples and the other method for all the other samples?

Line 172: Delete extra comma.

Line 190: “For pairwise comparisons of the loggerhead diatom flora from four localities, the Sørenson similarity index [36] was calculated.” should be “To make pair-wise comparison in between geographic location, we used the Sørenson similarity index”

Line 213: Include ‘diatom’ before taxa and delete ‘during the counts’.

Line 234: In my opinion, it is not appropriate in this context to refer to turtles as Green turtles or South African turtles. These turtles migrate between several different countries and so the term “sampled in Greece” is more accurate than “Greek turtle”.

Line 254-255: It is not clear what you are trying to say here.

Line 363: I think you mean exact richness of the population, or do you mean individual? It is not clear.

Line 366: “Will be” should be “would be”.

Line 364 – 367: This is a run-on sentence.

Line 373: I think you mean “predominantly” not “dominantly”.

Line 381: I would add ‘sampled so far’ to the end of this sentence as very few turtles have been sampled and so you cannot say this with confidence.

Line 380: Change “known” to “observed”.

Line 399: Run-on sentence.

Line 410: This makes it sound as if this was not found in non-Greek turtles, which were not sampled. Perhaps, rephrase to say “For Greek turtles, we sampled both skin and carapace. Interestingly, we found differences…”

Line 421: I am confused as to why you would expect to find obligate epibiont diatoms in the benthos. Surely, then they wouldn’t be obligate epibionts? Please explain.

Line 451: “As the biofilm accumulates, the available, uncolonized substratum surface on the carapace decreases and the relative abundance of strictly epizoic diatom taxa that needs this kind of substratum to thrive, declines” should be “As biofilm accumulates, the available and uncolonized substratum surface on the carapace decreases and so there will also be a decline in the relative abundance of strictly epizoic diatom taxa”.

Line 455: Two double spaces in the sentence.

Line 458: Double space in this sentence.

Line 460: Wandering is a very informal and inaccurate word in this context. “Satellite tracking of loggerheads showed that Amvrakikos Gulf turtles display long-term residency in this area, occasionaly wandering off to the northern Adriatic Sea using the latter as a foraging ground” should be “Satellite tracking have revealed that loggerhead turtles in Amvrakikos Gulf generally remain resident in this area but do occasionall venture to the northern Adriatic sea to forage”.

Line 490: Satellite tracking would be a more appropriate term than GPS tracking.

Reviewer #2: The authors made most of the suggested changes and presented acceptable reasons for suggestions that did not result in changes.

I found only two minor grammatical changes that are needed:

1. In the last sentence of the Abstract, 'between' should be 'among' because more than two items are being compared.

2. on line 480, 'numberof' should be 'number of'

7. PLOS authors have the option to publish the peer review history of their article (what does this mean?). If published, this will include your full peer review and any attached files.

Reviewer #1: Yes: Nathan J. Robinson

Reviewer #2: Yes: Elizabeth Bergey

---

## [Author Response · Author response to Decision Letter 1]

12 Jun 2020

PONE-D-20-05852

Diversity of diatom communities associated with loggerhead sea turtles (Carreta carreta)

PLOS ONE

Response to reviewers

Dear Editor,

Thank you and the two reviewers for their comments and suggestions. We made the requested changes, both in the text and in the figures, please find our answer (A) to the reviewers’ suggestions (R) below. 

R: I still agree that the study presented by Van de Vijver et al. is interesting and worthy of publication. Nevertheless, on a detailed read through, I have noted several typos, errors, and situations were the language could be improved. Thus, I think that the text needs some considerable improvements to be at the level where it would be ready for publication. I have listed several comments below. On top of this, I would also recommend that the authors also take the time to give this manuscript a detailed read through before resubmission to make sure that the text is as clear and understandable as possible.

A: Thank you for your comments and suggestions. We have checked the manuscript text and corrected it where we found mistakes or unclear wording. 

R: Title: “Carreta caretta” should be “caretta caretta”.

A: Corrected

R: Line 25: I would delete this first sentence. The second sentence is a perfect location for this manuscript to start.

A: We deleted the first sentence of the abstract and now it starts with the sentence: “Epizoic diatoms form an important part …”

R: Line 28: “skins and shells” do not need to be pluralized in this context.

A: Corrected

R: Line 28: Please include the scientific names after you first provide the species common name.

A: We added the scientific name: „loggerhead sea turtles (Caretta caretta)”

R: Line 33: “While sampling technique differed between locations, and this may affect some of variations we observed between localities, we believe that geography played a larger role” should be “We believe that part of this variation was due to differences in sampling techniques; however, we still believe that geography had an important role.”

A: We have changed this sentence to: „Part of this variation could be attributed to differences in sampling techniques; however, we believe that geography had an important role.”

R: Line 35 – 37: “Only 5 000 of the known diatom species are considered to be marine with 50 another 50 000 diatom species still to be discovered and described in the marine realm” should be “Of these, around 55,000 are estimated to exist in marine habitats. To date however, less than 5,000 of marine diatoms have been described.”

A: The sentence was changed according to the reviewer suggestion.

R: Line 51-52: This sentence should be moved to be the second sentence in this paragraph.

A: We did not move this sentence as this is the start of the description of diatom habitats, and that is also decribed in the sentences that follow to the end of the paragraph. 

R: Line 62: You already mention that diatoms are found on whales and sea turtles in Line 57. Thus, it is a little confusing to again mention all the different species that diatoms are found on just a few sentences later. I would delete Lines 62 – 65 and start by talking about sea turtles.

A: The first sentence illustrates examples of marine vertebrate hosts for epizoic diatom, and the second sentence is listing some of the taxa, only mentioning their host (cetaceans). We feel that the deleting the suggested lines would influence the context of previously known information listed in the Introduction. 

R: Line 70: Delete “being”.

A. Deleted

R: Line 82: This should be written in the past not present tense.

A: Changed to the past tense.

R: Line 111 – 112: I would argue that this is unnecessary and should be deleted.

A. The suggested sentence was deleted.

R: Line 115: I do not think that originates is the appropriate word here. The population does not breed in Amvrakikos, it just forages here.

A: We changed the expression „originates“ to „was sampled in Amvrakikos Gulf“.

R: Line 128: “All samples were collected in a non-invasive way for the animal.” I would delete this sentence as you explain this in the methods.

A: The sentence has been removed.

R: Line 150: “In total, 25 samples were analyzed, twenty from the carapace and five from the skin.” This is a very confusing way to write this. I would change it to “In total, we collected 25 samples from loggerhead skin. In addition, for five of these animals we were also able to simultaneously sample their carapaces for diatoms.”

A: We wrote: „In total, we collected 25 samples of which 20 from loggerhead carapace and for five of these animals we were also able to simultaneously sample their skin.”

R: Table 1: This could be condensed as much information is repeated. How about one line for each sampling location with ranges?

A: We have condensed the information in the Table 1, and added the range of geographical coordinates to the location name as suggested.

R: Line 159-160: “Samples were processed following the methods described by Hasle and Syvertsen [30] for South African samples and van der Werff [31].” This is a confusing way to put this. Are you trying to say that you used the Hasle and Syvertsen method only for South African samples and the other method for all the other samples?

A: Yes, that is right. The methods are further described in the same paragraph. We added the appropriate citation after each sentence where we specify the method used.

Line 172: Delete extra comma.

A: Deleted

Line 190: “For pairwise comparisons of the loggerhead diatom flora from four localities, the Sørenson similarity index [36] was calculated.” should be “To make pair-wise comparison in between geographic location, we used the Sørenson similarity index”

A: Corected.

R: Line 213: Include ‘diatom’ before taxa and delete ‘during the counts’.

A: We added diatom, but did not delete during the counts as besides the counting, we afterwards identified additionally 214 taxa, as stated further in the paragraph.

R: Line 234: In my opinion, it is not appropriate in this context to refer to turtles as Green turtles or South African turtles. These turtles migrate between several different countries and so the term “sampled in Greece” is more accurate than “Greek turtle”.

A: Yes, we agree with the reviewer and we refer to the samples as Greek, Croatian etc.. and to turtles as sampled in Greece, Croatia etc..

R: Line 254-255: It is not clear what you are trying to say here.

A: We have changed the sentence to: Although most taxa occurred in only one investigated locality…

R: Line 363: I think you mean exact richness of the population, or do you mean individual? It is not clear.

A: Taxon richness found in all epizoic samples examined in the study. 

R: Line 366: “Will be” should be “would be”.

A: Corrected.

R: Line 364 – 367: This is a run-on sentence.

A: the sentence was split into two parts, and now says: 

“Additionally, several taxa mostly belonging to the genera Amphora, Navicula and Nitzschia were grouped under a common name and detailed SEM and molecular analysis would be necessary to clarify their correct taxonomic identity. That would most likely result in the increase of the true taxa richness.”

R: Line 373: I think you mean “predominantly” not “dominantly”.

A: Yes, thank you

R: Line 381: I would add ‘sampled so far’ to the end of this sentence as very few turtles have been sampled and so you cannot say this with confidence.

A: Added.

R: Line 380: Change “known” to “observed”.

A: Changed.

R: Line 399: Run-on sentence.

A: The sentence was rewritten to: “Despite the high taxon richness, the percentage of the presumably truly epizoic taxa is rather low, although, we cannot be certain of an exact number of taxa that belong to that group.”

R: Line 410: This makes it sound as if this was not found in non-Greek turtles, which were not sampled. Perhaps, rephrase to say “For Greek turtles, we sampled both skin and carapace. Interestingly, we found differences…”

A: We rephrased this sentences as reviewer suggested.

R: Line 421: I am confused as to why you would expect to find obligate epibiont diatoms in the benthos. Surely, then they wouldn’t be obligate epibionts? Please explain.

A: Yes, we cannot exclude the possibility that some of the taxa that we consider as epizoic canot thrive on other surfaces. Marine benthic diatom diversity is largely unexplored and new discoveries are more than possible. However, in this part we simply stated that in some diatom genera there are species that are epizoic and some species that are not epizoic, but commonly found in benthic habitats. 

R: Line 451: “As the biofilm accumulates, the available, uncolonized substratum surface on the carapace decreases and the relative abundance of strictly epizoic diatom taxa that needs this kind of substratum to thrive, declines” should be “As biofilm accumulates, the available and uncolonized substratum surface on the carapace decreases and so there will also be a decline in the relative abundance of strictly epizoic diatom taxa”.

A: We rephrased the sentence as reviewer suggested.

Line 455: Two double spaces in the sentence.

A: Corrected.

Line 458: Double space in this sentence.

A: Corrected.

Line 460: Wandering is a very informal and inaccurate word in this context. “Satellite tracking of loggerheads showed that Amvrakikos Gulf turtles display long-term residency in this area, occasionally wandering off to the northern Adriatic Sea using the latter as a foraging ground” should be “Satellite tracking have revealed that loggerhead turtles in Amvrakikos Gulf generally remain resident in this area but do occasionally venture to the northern Adriatic sea to forage”.

A: We rephrased the sentence as reviewer suggested.

Line 490: Satellite tracking would be a more appropriate term than GPS tracking.

A: Corrected.

Reviewer #2: The authors made most of the suggested changes and presented acceptable reasons for suggestions that did not result in changes.

I found only two minor grammatical changes that are needed:

1. In the last sentence of the Abstract, 'between' should be 'among' because more than two items are being compared.

2. on line 480, 'numberof' should be 'number of'

A: Both mistakes were corrected.

---

## [Decision Letter · Decision Letter 2]

9 Jul 2020

Geographical variation in the diatom communities associated with loggerhead sea turtles (Caretta caretta)

PONE-D-20-05852R2

Dear Dr. Bosak,

We’re pleased to inform you that your manuscript has been judged scientifically suitable for publication and will be formally accepted for publication once it meets all outstanding technical requirements.

Kind regards,

Vona Méléder, Ph.D.

Academic Editor

PLOS ONE

Additional Editor Comments (optional):

Reviewers' comments:

Reviewer's Responses to Questions

**Comments to the Author**

1. If the authors have adequately addressed your comments raised in a previous round of review and you feel that this manuscript is now acceptable for publication, you may indicate that here to bypass the “Comments to the Author” section, enter your conflict of interest statement in the “Confidential to Editor” section, and submit your "Accept" recommendation.

Reviewer #1: (No Response)

2. Is the manuscript technically sound, and do the data support the conclusions?

Reviewer #1: Yes

3. Has the statistical analysis been performed appropriately and rigorously? 

Reviewer #1: Yes

4. Have the authors made all data underlying the findings in their manuscript fully available?

Reviewer #1: Yes

5. Is the manuscript presented in an intelligible fashion and written in standard English?

Reviewer #1: Yes

7. PLOS authors have the option to publish the peer review history of their article (what does this mean?). If published, this will include your full peer review and any attached files.

Reviewer #1: **Yes: **Nathan Jack Robinson

---

## [Editor Report · Acceptance letter]

15 Jul 2020

PONE-D-20-05852R2 

Geographical variation in the diatom communities associated with loggerhead sea turtles (Caretta caretta) 

Dear Dr. Bosak:

I'm pleased to inform you that your manuscript has been deemed suitable for publication in PLOS ONE. Congratulations! Your manuscript is now with our production department. 

Kind regards, 

on behalf of

Dr. Vona Méléder 

Academic Editor

PLOS ONE